# Multiscale Non-stationary Causal Structure Learning from Time Series Data

## Abstract

We introduce a new type of causal structure, namely *multiscale non-stationary directed acyclic graph* (MN-DAG), that generalizes DAGs to the time-frequency domain. Our contribution is twofold. First, by leveraging results from spectral and causality theories, we expose a novel probabilistic generative model, which allows to sample an MN-DAG according to user-specified priors concerning the time-dependence and multiscale properties of the causal graph. Second, we devise a Bayesian method for the estimation of MN-DAGs, by means of *stochastic variational inference* (SVI), called Multiscale Non-Stationary Causal Structure Learner (MN-CASTLE). In addition to direct observations, MN-CASTLE exploits information from the decomposition of the total power spectrum of time series over different time resolutions. In our experiments, we first use the proposed model to generate synthetic data according to a latent MN-DAG, showing that the data generated reproduces well-known features of time series in different domains. Then we compare our learning method MN-CASTLE against baseline models on synthetic data generated with different multiscale and non-stationary settings, confirming the good performance of MN-CASTLE. Finally, we show some insights derived from the application of MN-CASTLE to study the causal structure of 7 global equity markets during the Covid-19 pandemic.

## 1 Introduction

A causal graph describes causal relationships among the constituents of a given system, and represents a powerful tool for analyzing such a system under interventions and distribution changes. In general, causal graphs are not known. Fortunately, it is possible to leverage causal inference approaches to unveil and quantify the causal relationships among variables. While randomized experiments are the gold standard for testing causal hypotheses (especially in medicine and social sciences), in many cases such interventional approaches are unfeasible or unethical. Therefore, in recent years, great effort has been devoted to the development of methods able to retrieve causal structures from observational data (Glymour et al., 2019; Schölkopf et al., 2021).

Regardless of the different causal inference methods, the most informative causal graph is a *directed acyclic graph* (DAG), where the nodes in $\mathcal{V}$ are the variables of the system, all edges $e_{ij} \in \mathcal{E} \subseteq \mathcal{V} \times \mathcal{V}$ are directed and represent direct causal effects, and feedback loops among nodes are forbidden (acyclicity requirement). A DAG can be associated with its functional representation, also known as *structural equation model* (SEM, Pearl 2009). Here each node of the causal graph is written as a function of the values of a set of parents nodes and of an endogenous latent noise (see 3.2). In this work, we focus on the case in which such functions are linear and the latent noise is additive.

Even though widely studied and applied, a linear SEM is not adequate to cope with causal relations that evolve over time and occur at different time scales, which are both common when dealing with time series. Indeed, a SEM assumes that (i) causal edges and their weights are stationary and (ii) there is only one time scale at which causal relations occur, i.e., the one associated with the frequency of observed data. However, in practice causal structures might be non-stationary (D'Acunto et al., 2021) and often there is no prior

knowledge about the temporal resolutions at which causal relations occur (Runge et al., 2019; D'Acunto et al., 2022).

To overcome these limits, we introduce multiscale non-stationary causal structures, namely MN-DAGs, that generalize linear DAGs to the time-frequency domain. In our work, the term *multiscale* means that we consider multiple time resolutions, i.e., frequency bands. Hence, we look for causal interactions among time series within each of those distinct frequency bands, and we simultaneously inspect the behaviour of these causal relationships along time. Throughout the paper, we use $2^j$ to represent a certain temporal resolution, where $j = \{1, \ldots, J\}$ indicates the associated scale level and $J \in \mathbb{N}$ is the maximum level considered.

In MN-DAGs each time scale is represented by a different graph page (akin to multi-layer networks). Then, the vertices within a certain page are associated with the multiscale representation of the $N$ time series at the frequency corresponding to that page. There exists a unique global causal ordering '$\prec$' shared by all graph pages, such that the possible parent set $\mathcal{P}_{i,\prec}$ for the $i$-th node $X_i$ can include only by those nodes $X_j$ that precede it in the causal ordering ($X_j \prec X_i$). Causal relationships among nodes, represented as directed edges, can vary smoothly over time and constitute acyclic structures within each time scale. So, throughout the paper the term non-stationarity associated with causal structures refers to a smooth dependence on time, similarly to how it is defined by Huang et al. (2020).

## 1.1 Contributions and Roadmap

**Probabilistic generative model.** As a first contribution, we propose a novel probabilistic generative model over MN-DAGs, having as latent variables the causal ordering and the causal relationships; while the observables are $N$ zero-mean time series of length $T$.

Our generative model combines multivariate locally stationary processes (MLSW, Park et al. 2014), a framework to represent time series as a sum of contributions coming from different temporal resolutions, and linear SEM. In particular, we establish a relation between the underlying causal structure and the *transfer function matrix*, that determines the local variance and cross-covariance between the time series (see 3.2). To this end, we replace the transfer function matrix with a time-dependent mixing matrix, determined by the causal structure and the strength of causal relations at each time step and scale level.

The proposed probabilistic model takes as input (i) the number of nodes $N \in \mathbb{N}$; (ii) the number of samples $T \in \mathbb{N}$; (iii) a parameter $\mu \in [0, 1]$ associated with the multiscale feature; (iv) $\tau \in [0, 1]$ which describes the time dependence of causal relationships; (v) $\delta \in [0, 1]$ that manages the density of the MN-DAG.

Figure 1 shows a $(\tau, \mu)$-quadrant along with examples of latent causal structures which determine the sampled data, according to the specified values of $\mu$ and $\tau$. When $\mu = 0$, we obtain the single-scale case depicted in Figure 1(a) and 1(b). Here, the MN-DAG has only one page. We assume that the power spectrum describing the system is concentrated in the finest scale level $j = 1$. Moreover, if also $\tau = 0$, the causal links are stationary (Figure 1(a)). Starting from the origin, as we move to the right ($\tau \to 1$) the temporal dependence of causal connections increases. As we move upwards ($\mu \to 1$), the likelihood that the causal graph contains more pages increases (Figure 1(c) and 1(d)). Then, the overall power spectrum is spread over more temporal resolutions.

Statistical analysis shows (see Section 4) that data generated by this model reproduces well-known features of real-world time series, such as serial correlation, weakly stationarity, volatility clustering, and non-normality. Our generative model can thus be used in practice to develop benchmarks for comparing causal discovery algorithms, whose performance can be measured as a function of $\mu$ and $\tau$.

Our probabilistic generative model is also suitable for practitioners, who need to identify the best performing models for their specific application domain. In fact, some underlying assumptions of causal inference models, such as, e.g., stationarity of causal relationships and distributional characteristics of latent factors, might be violated by real-world data coming from specific application domains. When this happens, it impairs the ability of existing methods to recover the underlying causal structure.

**Bayesian causal inference method.** Our second main contribution is a Bayesian method for causal structure learning, able to cope with multiscale data which features time-dependent variance. Our method,

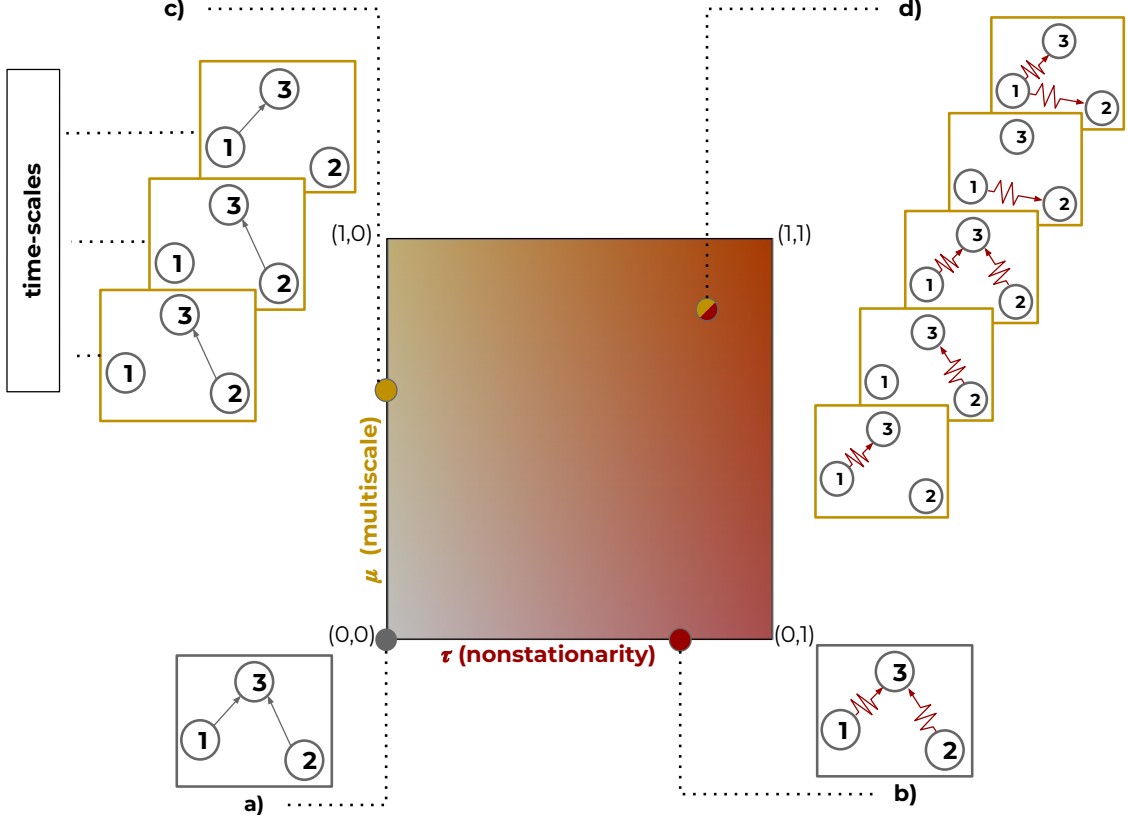

Figure 1: Examples of causal structures that can be sampled from the proposed probabilistic model according to the specified values for non-stationarity and multiscale features. In the depicted quadrant, we have the non-stationarity feature (associated with the parameter $\tau \in [0, 1]$) on the x-axis in red, and the multiscale feature (associated with the parameter $\mu \in [0, 1]$) on the y-axis in yellow, respectively. Colors, edges shape and number of graph layers highlight differences from the single-scale stationary DAG corresponding to the origin of the quadrant (a). When we move horizontally, the temporal dependence of the causal coefficients (edges in the causal graph) changes (b). Similarly, vertical shifts in the quadrant are associated to the change in the number of time scales (pages of the causal graph) contributing to the sampled data (c). Finally, when both $\tau$ and $\mu$ are different from zero, we sample data concerning a system driven by an underlying multiscale non-stationary causal structure (MN-DAG).

named MN-CASTLE, relies upon observational data and the estimate of the decomposition of the power spectrum at different temporal resolutions.

The main challenges in developing such a model are to (i) handle the variance of the gradient estimator of the *evidence lower bound* objective function with respect to the causal ordering, where the latter is modeled by a Plackett-Luce distribution; (ii) impose the causal ordering to optimize at each inference step only the (acyclic) causal relations that conform to it; and (iii) optimize smoothly time-dependent latent causal relations efficiently, while containing the number of parameters. Regarding the first point, since the causal ordering is non-reparametrizable (see Appendix A), we use a data-dependent control variate strategy along with the gradient estimator (see Section 3.3). Furthermore, in order to impose a given causal ordering, we generate a permutation tensor from the latter, and mask the distributions of the probabilistic model accordingly. Finally, we model the time-varying causal coefficients as latent batched Gaussian processes, all having the same kernel. Here, we exploit a variational formulation of the Gaussian processes in order to

reduce the computational burden of the estimation while avoiding overfitting at the same time. Experimental results on synthetic datasets, generated by using the proposed probabilistic model, show that MN-CASTLE is able to retrieve the latent causal structure for different configurations of $(\tau, \mu)$ values, and that it outperforms baseline models.

We then apply MN-CASTLE to study the causal structure of 7 global equity markets during the Covid-19 pandemic. Our findings indicate that the most relevant time scale is the finest one, which corresponds to a resolution of 2-4 days. This is in agreement with the presence of financial turbulence during the analyzed period, which led investors to react quickly to the shocks that followed, thus generating sudden swings in stock prices. In addition, our results on the causal ordering show that Asian markets are the main drivers of returns within the considered network. Then we find the European equity markets, and finally the U.S. stock market. So, the causal relationships tend to reflect the spread of the epidemic, with reference to the number of confirmed cases in different countries. In conclusion, we observe that the causal relationships tend to be positive and increase in intensity during the early months of 2020. Therefore, changes in value in Asian stock markets cause changes in value in European and U.S. stock markets, consistently with the presence of the pandemic and the subsequent fears of investors worldwide.

Summarizing, the technical contributions of this paper are as follows:

- We define a new type of causal structure for time series data (MN-DAG).
- We devise a probabilistic generative model, which allows to sample an MN-DAG according to user-specified priors concerning the time-dependence and multiscale properties of the domain. Our model can be used to generate synthetic time series with real-world characteristics.
- We devise a Bayesian inference method for the estimation of MN-DAG from real-world data (MN-CASTLE).

At a high level, this paper bridges the gap between multiscale modeling and machine learning-based causal inference methods.

This article is organized as follows. Section 2 relates our method to existing ones, highlighting differences and similarities. Then, Section 3 contains the novel methodological content of the paper. It is further split into three subsections, that show how to sample MN-DAG (Section 3.1); how to generate data from MN-DAG (Section 3.2); how to infer MN-DAGs from data (Section 3.3). Next, Section 4 provides the obtained results. In detail, Section 4.1 statistically describes data generated by the probabilistic generative model. Section 4.2 regards tests on synthetic datasets, by providing details concerning the experimental settings and introducing the considered baseline models. Section 4.3 analyses a real world use case on global equity markets. Finally, Section 5 concludes by giving additional discussion concerning our findings, outlines open questions and future research directions.

## 2    Related Work

Causal inference methods can be mainly classified into three categories, according to the approach used to infer the causal graph: (i) *constraint-based approaches*, which make use of conditional independence tests to establish the presence of a link between two variables (Spirtes et al., 2000; Huang et al., 2020); (ii) *score-based methods*, which use search procedures in order to optimize a certain score function (Heckerman et al., 1995; Chickering, 2002; Huang et al., 2018); (iii) *structural equation models*, which express a variable at a certain node as a function of its parents (Shimizu et al., 2006a; Hoyer et al., 2008; Shimizu et al., 2011; Peters et al., 2014; Bühlmann et al., 2014). Our approach fits into the latter category and aims to handle the presence of non-stationarity and different temporal resolutions in the underlying causal structure.

Unlike the multiscale causal structure learning model proposed by D'Acunto et al. (2022), which estimates multiscale stationary causal relationships hinging on stationary wavelet transform (Nason & Silverman, 1995) and non-convex optimization, our model applies a different learning scheme and is able to handle non-stationary relationships as well. Furthermore, the method we propose exploits the estimate of the

decomposition of the power spectrum at different time scales, whereas the model proposed in the previous paper operates on the estimated wavelet detail vectors.

In the past, several approaches have been developed that can infer causal structures in the presence of non-stationarity under certain assumptions (Song et al., 2009; Ghassami et al., 2018; Strobl, 2019; Perry et al., 2022). The main (implicit) assumption common to these approaches, concerns the time scale at which causal interactions occur, that is, it is assumed that this scale coincides with the frequency of observation of the data. The model we propose, relaxes this assumption, and allows time-dependent causal relationships to be investigated at different temporal resolutions. Another difference concerns the assumption regarding the existence of multiple domains, where causal dependencies between variables may vary but are assumed to be stationary within each domain, to exploit non-stationarity and distributional shifts to recover the underlying causal structure. Although in the context of time series, the dataset can be segmented into different domains through a sliding window approach, this procedure introduces discretionary choices such as (i) the choice of the splitting points and (ii) the size of the time window in which causal relationships should be stationary. However, in general, for real data there is no prior knowledge regarding the above issues: the causal structure might vary a lot even when windows are overlapping (D'Acunto et al., 2021). In contrast, our method aims to learn the causal structure and describe its temporal evolution, assuming that it is linear in the frequency domain and that the causal ordering is shared between the temporal resolutions considered.

Our probabilistic generative model extends the works of Cundy et al. (2021); Charpentier et al. (2022), since it is suitable for time-series data and provides a causal structure that lives in the time-frequency domain. Even though our approach leverage Gumbel distributed variables for sampling the causal ordering as in the previous two works, the procedure we apply is different and requires a lower computational cost (Gadetsky et al., 2020). Moreover, our inference model uses a gradient estimator with a data-dependent control variate strategy for learning the parameters of the causal ordering distribution, whereas existing models exploit differentiable relaxations of such a distribution. Our procedure uses the masking of distributions as well to optimize at each step only the causal relations compliant to a certain causal ordering. A similar approach is also employed in Ke et al. (2019); Ng et al. (2022). however the masking used in those works aims at excluding all non-causal relations, not just those that do not conform to the causal ordering.

Finally, we exploit recent developments in variational inference in order to approximate the posterior distribution over MN-DAG parameters given data, in accordance with the MN-CASTLE probabilistic model. This general learning scheme is also exploited in other recent works (Cundy et al., 2021; Charpentier et al., 2022; Annadani et al., 2021; Lorch et al., 2022) to model the posterior distribution over the parameters of a DAG, as defined in the corresponding proposed probabilistic models.

## 3   Methods

This section discusses the proposed generative model and inference method in detail. Specifically, Section 3.1 details the construction of the probabilistic model useful for sampling MN-DAG. Then, Section 3.2 defines the data generation process from an MN-DAG, which combines MLSW and SEM. Finally, Section 3.3 analyses MN-CASTLE and the inference procedure used.

### 3.1   Sampling a MN-DAG

We propose a probabilistic model over MN-DAGs that takes as input (i) the number of nodes $N \in \mathbb{N}$; (ii) the number of samples $T \in \mathbb{N}$; (iii) a variable $\mu \in [0, 1]$ associated with the multiscale nature of the DAG; (iv) $\tau \in [0, 1]$ which describes the time dependence of causal relationships; (v) $\delta \in [0, 1]$ that describes the density of the causal network. Figure 2 shows the steps needed to sample an MN-DAG.

**Sample the time scales.** Given $\mu$, the number of pages (time scales) of the MN-DAG is $J = 1 + J'$, where $J'$ is sampled from a binomial distribution $J' \sim B(\log_2(T) - 1, \mu)$. Here, the first parameter of the binomial distribution is the number of trials and the $\mu$ represent the probability of success. Without loss of generality, we assume that temporal resolutions are consecutive, i.e., given the value of $J$, all the time scales $2^j$, $j = \{1, \ldots, J\}$, are associated with a page in the causal graph. This assumption does not imply that causal relations occur within all the considered pages. Since the model is probabilistic and the user specifies

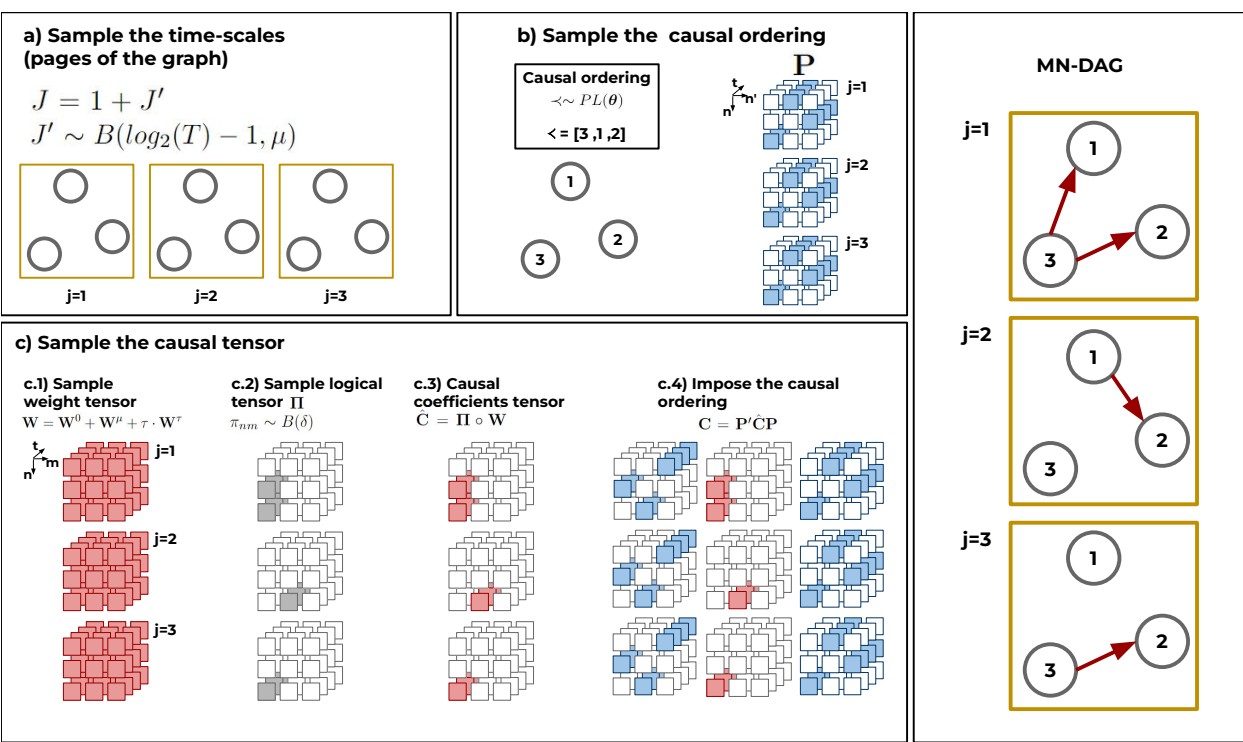

Figure 2: The figure shows the steps necessary to sample a MN-DAG. For the sake of readability, let us consider the case where N=3 and T=4. Here the yellow color refers to time scales; red for non-stationarity; (a) First, we sample the number of pages (time scales) of the MN-DAG. Given $\mu$, the latter is given by $J = 1 + J'$, where $J' \sim B(log_2(T) - 1, \mu)$. In the example, we instantiate three pages ($J = 3$). (b) Second, we sample the causal ordering $\prec \sim PL(\boldsymbol{\theta})$ that is shared by all time scales and entails the permutation tensor $\mathbf{P} \in \{0, 1\}^{J \times T \times N \times N}$. Here, $PL(\boldsymbol{\theta})$ indicates the Plackett-Luce distribution, defined by a score vector $\boldsymbol{\theta} \in \mathbb{R}^N$, where $\theta_i \sim U(0, N)$. The indexes are $j$ for time scales, $t$ for time steps, $n$ for the considered nodes and $n'$ for the positions within $\prec$. In the considered example, we have $\prec = [3, 2, 1]'$. Therefore, for each $3 \times 3$ slice of the tensor corresponding to a certain time scale $j$ and time $t$, we have $p_{nn'} = 1$ (blue square) if the node $n$ appears at index $n'$ within $\prec$. (c) Third, we build the tensor of causal coefficients as follows. With regards indexes, $j$, $t$ and $n$ are the same as above, whereas $m$ indicates the parents dimension. Here, we first sample a full tensor of weights $\mathbf{W} \in \mathbb{R}^{J \times T \times N \times N}$ made by three components: (i) a constant term $\mathbf{W}^0$; (ii) $\mathbf{W}^\mu$ that makes the magnitude of causal relationships different across time scales; (iii) $\mathbf{W}^\tau$ that allows causal coefficients to vary over time according to batched $GP(0, \mathbf{K})$. Therefore, within each scale $j$, $W_{nm}$ are smooth functions varying over index $t$. To manage the density of the entailed MN-DAG, we multiply element-wise $\mathbf{W}$ by a logical mask $\boldsymbol{\Pi} \in \{0, 1\}^{J \times T \times N \times N}$. The entries of the $\boldsymbol{\Pi}$ are distributed according to a Bernoulli distribution, $\pi_{nm} \sim B(\delta)$. Finally, we obtain the causal tensor $\mathbf{C}$ that entails the MN-DAG on the right by imposing the causal ordering sampled at step (b).

a value for $\delta$, we also might end up with a causal graph without edges. In case $\mu = 0$, the causal graph consists of one page, that corresponds to the time resolution $2^1$, i.e., single time steps.

**Sample the causal ordering.** Within our probabilistic model, we assume that the causal ordering $\prec$ is shared by all time scales. This property implies that, given $\prec$, the possible parent sets at each temporal resolutions $\mathcal{P}_{i,\prec}$ for the $i$-th variable $X_i$ are $\{\mathcal{P}_i \mid \mathcal{P}_i \prec X_i\}$. The causal ordering $\prec$ can be thought as a permutation of a vector of integers $\prec' = [1, \ldots, N]$, thus we exploit the *Plackett-Luce* distribution (PL, Plackett 1975; Luce 2012) to sample it. $PL$ represents a distribution over permutations, defined by a vector of scores $\boldsymbol{\theta} \in \mathbb{R}^N$, which allows sampling permutations $\mathbf{b} \in \mathcal{S}_N$ in $\mathcal{O}(N \log N)$, where $\mathbf{b}$ is a vector of $N$ integers and $\mathcal{S}_N$ is the support of permutations of $N$ elements. Thus, given $\boldsymbol{\theta}$, the probability of a

permutation $\mathbf{b}$ is

$$p(b \mid \theta) = \prod_{i=1}^{k} \frac{e^{\theta_{b_i}}}{\sum_{u=i}^{k} e^{\theta_{b_u}}} .$$

A sample $\mathbf{b}$ from $PL$ distribution can be thought as a sequence of samples from categorical distributions: first $b_1$ comes from the categorical distribution with logits $\boldsymbol{\theta}$; $b_2$ from the categorical with logits $\boldsymbol{\theta} - \{\theta_{b_1}\}$; and so on. The mode of the $PL$ is the descending order permutation of scores $\mathbf{b}^0 = \theta_{b_1^0} \geq \theta_{b_2^0} \geq \ldots \geq \theta_{b_N^0}$. The sampling procedure from a $PL$ relies upon the fact that an order of a vector $\mathbf{z} \in \mathbb{R}^N \sim Gum(\boldsymbol{\theta}, 1)$ is distributed as $PL(\boldsymbol{\theta})$, where $Gum(\boldsymbol{\theta}, 1)$ is a Gumbel distribution with location parameter $\boldsymbol{\theta}$ and scale equal to one (Gadetsky et al., 2020). Therefore, we can sample $\mathbf{b}$ as follows:

$$z_i = \theta_i - \log(-\log(v_i)), \qquad v_i \sim U(0, 1)$$
$$H(\mathbf{z}) = \operatorname{argsort}(\mathbf{z}) .$$

We sample the causal ordering $\prec \sim PL(\boldsymbol{\theta})$ by using the procedure above, where we choose a uniform prior for $PL$ scores vector, i.e., $\theta_i \sim U(0, N)$. The causal ordering $\prec$ entails a permutation matrix $\widehat{\mathbf{P}} \in \{0, 1\}^{N \times N}$ such that $p_{nn'} = 1$ iff the variable $X_n$ occurs at position $n'$ within $\prec$; 0 otherwise. Finally, we derive a permutation tensor $\mathbf{P} \in \mathbb{R}^{J \times T \times N \times N}$ by simply tiling $\widehat{\mathbf{P}}$ along both multiscale and time dimension.

**Sample the causal tensor.** Given $J$ and $\tau$, we build a tensor of weights $\mathbf{W} \in \mathbb{R}^{J \times T \times N \times N}$ made by three building blocks. First, we sample $\widehat{\mathbf{W}}^0 \in \mathbb{R}^{N \times N}$, whose entries are normally distributed, $w_{nm}^0 \sim N(0, 1)$. Starting from $\widehat{\mathbf{W}}^0$, we derive the first component $\mathbf{W}^0 \in \mathbb{R}^{J \times T \times N \times N}$ by simply expanding $\widehat{\mathbf{W}}^0$ along both multiscale and time dimension. Then, this component can be thought as a constant term shared by all temporal resolutions and time steps.

Second, we sample $\widehat{\mathbf{W}}^\mu \in \mathbb{R}^{J \times 1 \times N \times N}$, whose entries are distributed according a Gaussian $N(0, \mu)$. By expanding $\widehat{\mathbf{W}}^\mu$ along time dimension, we obtain the second component $\mathbf{W}^\mu \in \mathbb{R}^{J \times T \times N \times N}$, that makes the magnitude of causal relationships different across scales and is stationary along $t$.

Third, we sample $\mathbf{W}^\tau \in \mathbb{R}^{J \times T \times N \times N}$ where each tube along the time dimension follows a multivariate Gaussian distribution $MN(\mathbf{0}, \mathbf{K})$. Here, the covariance matrix $\mathbf{K} = \mathbf{K}(t, t')$ represents a (combination of) valid kernel(s) for Gaussian processes (GP, Bishop & Nasrabadi 2006), where the lengthscale is $\lambda = 1/\tau$. This component imposes the causal coefficient to evolve smoothly over time, according to $\tau$. Indeed, as $\tau \to 0$, the lengthscale of the kernel increases and consequently $\mathbf{W}^\tau$ varies less along the time dimension. Finally, the tensor of weights is

$$\mathbf{W} = \mathbf{W}^0 + \mathbf{W}^\mu + \tau \cdot \mathbf{W}^\tau . \tag{1}$$

Now, to manage the sparsity and ensure the acyclicity of causal connections, we generate a suitable logical mask $\widehat{\mathbf{\Pi}} \in \{0, 1\}^{J \times 1 \times N \times N}$. Within the latter, the slices $\widehat{\mathbf{\Pi}}_{nm}$ are strictly lower triangular and the entries are distributed according to a Bernoulli distribution, $\pi_{nm} \sim B(\delta)$. Then, we obtain the tensor of causal relations as $\widehat{\mathbf{C}} = \mathbf{\Pi} \circ \mathbf{W}$, whose slices $\widehat{\mathbf{C}}_{nm}$ are nilpotent.[1] Here $\mathbf{\Pi} \in \{0, 1\}^{J \times T \times N \times N}$ is obtained by expanding $\widehat{\mathbf{\Pi}}$ over time and $\circ$ represents the Hadamard product.

At this point, given $\mathbf{P}$ and $\widehat{\mathbf{C}}$, we compute the causal tensor that entails the latent MN-DAG by means of the product $\mathbf{C} = \mathbf{P}' \widehat{\mathbf{C}} \mathbf{P}$, where $\mathbf{P}'$ is obtained by transposing the two rightmost dimensions of $\mathbf{P}$.

### 3.2 Generate Data from the MN-DAG

In order to generate $N$ zero-mean processes of length $T$, whose behaviour is determined by the evolution over time of a latent MN-DAG, we build upon SEM and MLSW theoretical frameworks.

**Linear structural equation models and DAGs.** Mathematically, a DAG is formulated as a SEM. Given a dataset $\mathcal{X} := (x_1, \ldots, x_N)$ of $N$ random variables, a SEM is a collection of $N$ structural assignments

$$x_i := f_i(\mathcal{P}_i, z_i), \quad i = 1, \ldots, N,$$

---

[1] A matrix $\mathbf{A}$ is said nilpotent if it is square and $\mathbf{A}^{\bar{n}} = 0$ for all integers $\bar{n} \geq \bar{N}$, where $\bar{N}$ is known as the index of $\mathbf{A}$.

where $\mathcal{P}_i$ represents the set of direct causes (parents) of node $x_i$, $z_i$ is a noise variable satisfying $z_i \perp z_j$ if $j \neq i$, and $f_i(\cdot)$ is a generic functional form. In this paper, we focus on linear functional forms, therefore, by exploiting matrix form, the equation above becomes:

$$\mathbf{X} = \mathbf{CX} + \mathbf{Z},$$

where $\mathbf{C} \in \mathbb{R}^{N \times N}$ is the matrix of causal coefficients satisfying (i) $c_{ii} = 0 \quad \forall i \in \{1, \ldots, N\}$; (ii) $c_{ij} \neq 0 \iff x_j \in \mathcal{P}_i$; (iii) $diag(\mathbf{C}^n) = \mathbf{0}, \quad \forall n \in \mathbb{N}$ (acyclicity property). Since $\mathbb{I} - \mathbf{C}$ is an invertible matrix (see Appendix B), we can rewrite the latter equation as

$$\mathbf{X} = \mathbf{MZ}, \tag{2}$$

with $\mathbf{M} = (\mathbb{I} - \mathbf{C})^{-1}$ being a mixing matrix. According to Equation (2), observed data is a mixing of independent latent noises. Here, causal relations are stationary, instantaneous and are supposed to occur at the frequency of observed data.

**Modeling multiscale data.** In order to deal with multiscale data featured by time-dependent variance, it is necessary to leverage a mathematical modeling able to localize the evolution of the considered processes in time and frequency. Without loss of generality, let us consider zero mean processes (any non-zero mean can be estimated and removed).

The MLSW framework generalizes *locally stationary wavelet* process (LSW, Nason et al. 2000; see Appendix D) to model $N$ zero-mean processes $\mathbf{X}_T[t] = [X_1[t], \ldots, X_N[t]]'$, each of length $T$, as follows:

$$\mathbf{X}_T[t] = \sum_{j=1}^{J} \sum_{k=-\infty}^{+\infty} \mathbf{V}_j[k/T] \mathbf{z}_{j,k} \psi_j[t-k]. \tag{3}$$

In Equation (3), (i) $\{\psi_j[t-k]\}$ is a set of non-decimated wavelets; (ii) $\{\mathbf{z}_{j,k}\}$ is a set of random vectors $\mathbf{z}_{j,k} \sim N(\mathbf{0}, \mathbb{I}_{N \times N})$; (iii) $\mathbf{V}_j[k/T] \in \mathbb{R}^{N \times N}$ is the transfer function matrix, assumed to be lower triangular and with entries being Lipschitz continuous functions associated with Lipschitz constants $L_j^{(n,m)}$, $n \in \{1, \ldots, N\}$, $m \in \{1, \ldots, N\}$, such that $\sum_j L_j^{(n,m)} < \infty$. Local stationarity means that the statistical properties of the process vary slowly over time. This feature is essential in order to make learning possible (Nason et al., 2000), and within MLSW coincides with the Lipschitzianity assumption above. Here, the transfer function matrix $\mathbf{V}_j[\nu]$, with $\nu = k/T$ being the rescaled time (Dahlhaus, 1997), provides a measure of the local variance and cross-covariance between the processes at a certain time $\nu$ and scale $j$, i.e., the *local wavelet spectral matrix* (LWSM) $\mathbf{S}_j[\nu] = \mathbf{V}_j[\nu] \mathbf{V}_j'[\nu]$.

By construction, LWSM is symmetric and positive at each time $\nu$ and scale $j$. Within LWSM, diagonal elements $S_{nn}^j[\nu]$ represents the spectra of of the processes, whereas $S_{nm}^j[\nu]$ provides the cross-spectra between them. In addition, the local auto and cross-covariance functions, namely $c_{nn}(\nu, l)$ and $c_{nm}(\nu, l)$ (with $l$ being a certain lag), admit a formulation in terms of the LWSM (see Park et al. 2014 for further details).

**The proposed data generation process.** In order to develop our generative model, we leverage Equations (2) and (3). In particular, we establish a relationship between the causal tensor entailing the latent MN-DAG and the transfer function matrix. To this end, we replace the lower triangular $\mathbf{V}_j[\nu]$ in Equation (3) with a time-dependent mixing matrix $\mathbf{M}_j[\nu] = (\mathbb{I} - \mathbf{C}_j[\nu])^{-1}$, which is a permutation of a lower triangular matrix (see Appendix C), where $\mathbf{C}_j[\nu] \in \mathbb{R}^{N \times N}$ is the matrix of causal coefficients at time $\nu$ and scale $j$ described in Section 3.1. Since for every real-valued invertible matrix $\mathbf{A}$, the Gramian $\mathbf{AA}'$ is positive definite, we can use the previous mixing matrix to define $\mathbf{S}_j[\nu]$. Therefore, the aforementioned processes are given by

$$\mathbf{X}_T[t] = \sum_{j=1}^{J} \sum_{k=-\infty}^{+\infty} \mathbf{M}_j[k/T] \mathbf{z}_{j,k} \psi_j[t-k]. \tag{4}$$

In case (i) $\tau = 0$ (stationary $\mathbf{C}_j[\nu] = \mathbf{C}_j$) and (ii) either $\mu = 0$ ($J = 1$) or $\mathbf{C}_j = \mathbf{C}$ (i.e., causal coefficients do not vary over index $j$), Equation (4) simplifies as follows:

$$\mathbf{X}_T[t] = \mathbf{M} \sum_{j=1}^{J} \sum_{k=-\infty}^{+\infty} \mathbf{z}_{j,k} \psi_j[t-k] = \mathbf{M}\widetilde{\mathbf{Z}}[t] \, .$$

Although this representation looks similar to Equation (2), here we have that $\widetilde{\mathbf{Z}}[t]$ admits a representation of the form in Equation (3), with transfer matrix equal to the identity. Therefore, $\widetilde{\mathbf{Z}}[t]$ displays autocorrelation determined by the discrete autocorrelation wavelet $\mathbf{\Psi}_j[l] = \sum_k \psi_{j,k} \psi_{j,k-l}$, where $l \in \mathbb{Z}$ represents the lag (Park et al., 2014).

### 3.3 Alternating 2-Steps Inference

We expose a Bayesian method for the estimation of MN-DAGs, termed MN-CASTLE. The latter has been implemented by using Pyro (Bingham et al., 2019) a probabilistic programming language built on Python and PyTorch (Paszke et al., 2019). A probabilistic model is a stochastic function that generates data $\mathbf{x}$ according to latent random variables $\mathbf{z}$ and parameters $\boldsymbol{\beta}^*$, having as joint density function

$$p_{\boldsymbol{\beta}^*}(\mathbf{x}, \mathbf{z}) = p_{\boldsymbol{\beta}^*}(\mathbf{x} \mid \mathbf{z})p_{\boldsymbol{\beta}^*}(\mathbf{z}) \, ,$$

where $p_{\boldsymbol{\beta}^*}(\mathbf{z})$ and $p_{\boldsymbol{\beta}^*}(\mathbf{x} \mid \mathbf{z})$ are the prior and the likelihood, respectively. The goal is to learn the parameters of the model $\boldsymbol{\beta}^*$ from data. As detailed is Appendix A, SVI offers a scheme to learn $\boldsymbol{\beta}^*$ by approximating the usually intractable posterior distribution $p_{\boldsymbol{\beta}^*}(\mathbf{z} \mid \mathbf{x})$ by means of a tractable family of variational distributions $q_{\boldsymbol{\phi}}(\mathbf{z})$, here called guides, parameterized by the variational parameters $\boldsymbol{\phi}$.

Our task is as follows. We are given a dataset $\mathcal{X} = \{\mathbf{X}_T[t]\}_{t=1}^{T}$, $\mathbf{X}_T[t] = [X_T^1[t], \ldots, X_T^N[t]]'$ and an estimate of the LWSM at different time scales $j$, $\widehat{\mathbf{S}}_j$. As an example, the smoothed bias-corrected raw wavelet periodogram is a suitable non-parametric estimator (Park et al., 2014). Then, according to the probabilistic generative model in Section 3.2, we want to learn the following parameters given previous inputs by means of SVI: (i) the vector of scores $\boldsymbol{\theta}$ of the Plackett-Luce distribution used to model the latent global causal ordering $\prec$; (ii) the mean and kernel parameters of the latent batched GPs used to model the entries of the hidden causal coefficients tensor $\mathbf{C}$, i.e., $C_j^{(n,m)} \sim GP(\bar{C}_j^{(n,m)}, \mathbf{K}(t,t'))$. Here, we assume that the kernel $\mathbf{K}(t,t')$ is shared by all causal coefficients. Moreover, by learning the kernel parameters, we obtain an estimate $\hat{\tau}$ of $\tau$ since we assume $\tau = 1/\lambda$ as in Section 3.1.

---

**Algorithm 1** Training function

---

1: **procedure** TRAIN($\mathbf{X}$, $\widehat{\mathbf{S}}$, timesteps, $\mathbf{K}$)
2:     $model1, guide1 \leftarrow$ probabilistic model and guide for the first step
3:     $model2, guide2 \leftarrow$ probabilistic model and guide for the second step
4:     Initialize $niterations, optimizer$
5:     $SVI1 \leftarrow$ instantiate SVI object with $model1, guide1, optimizer$
6:     $SVI2 \leftarrow$ instantiate SVI object with $model2, guide2, optimizer$

7:     **while** i$<$ niterations **do**
8:         $\hat{\boldsymbol{\theta}}_i \leftarrow$ Run $i$-th $SVI1$ step
9:         $\hat{\prec}_i^0 \leftarrow$ sort $\boldsymbol{\theta}_i$ in descending order and retain the indexes
10:        $\widehat{\widehat{\mathbf{C}}}_i, \hat{\tau}_i \leftarrow$ Run $i$-th $SVI2$ step using the mode of $PL(\hat{\boldsymbol{\theta}}_i)$, $\hat{\prec}_i^0$
11:     **return** $\hat{\boldsymbol{\theta}}_i, \widehat{\widehat{\mathbf{C}}}_i, \hat{\tau}_i$

---

In order to improve the learning of the causal coefficients tensor, we split the inference of the causal ordering and the rest of the parameters into two alternating steps. Algorithm 1 shows the overall training function.

**Model and guide for causal ordering inference.** Figures 3a and 3b provide a pictorial representation of the probabilistic model and the guide initialized at line 2 of Algorithm 1. In particular, we resort to graphical models to illustrate the corresponding joint distributions. Here, random variables are given as circular nodes,

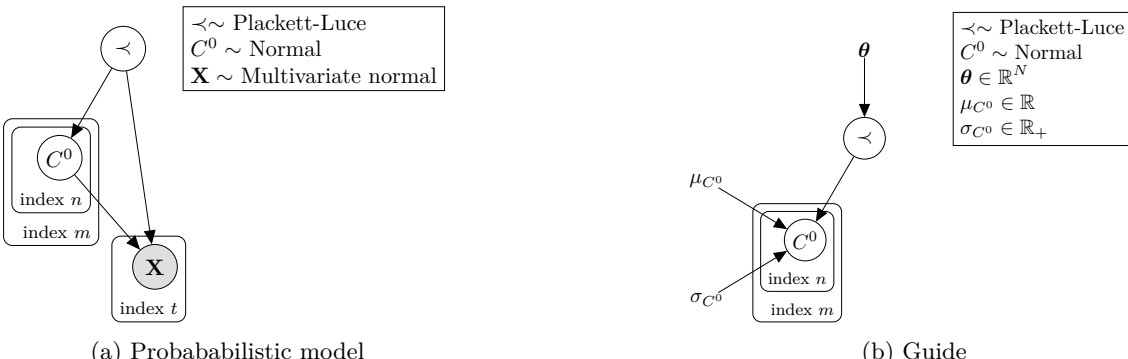

(a) Probababilistic model

(b) Guide

Figure 3: a Graphical models associated with the probabilistic model and b the parameterized variational distribution for learning the causal ordering, along with variational parameters and their constraints.

where a blank node represents a latent variable while a grey one is associated with an observed variable. Deterministic variables are given as rhomboid nodes, while variational parameters are printed outside of the nodes. Edges indicate dependence among variables and rectangles (plates) indicate conditionally independent dimensions, i.e., independent copies. In addition, Figure 3 provides the distributions (along with the constraints of parameters) of random variables and variational parameters.

Figure 3 shows that model and guide share the same latent variables. Indeed, since the guide is used to approximate the true posterior, it needs to provide a valid probability density over all hidden variables. More in detail, we have two latent variables: (i) the causal ordering, which is global since it does not depend on any other variable and is modeled within the guide as a $PL(\boldsymbol{\theta})$; (ii) a stationary single-scale causal structure $\mathbf{C}^0 \in \mathbb{R}^{N \times N}$, where each entry $C^0_{nm}$ is independent of the others and is modeled in the guide with a Gaussian. Since we assume the causal ordering (i) shared by all time scales and (ii) stationary; we infer it from observed data $\mathbf{X}$, without any additional information concerning the variance decomposition and its evolution over different temporal resolutions (provided by $\widehat{\mathbf{S}}_j$). For this reason, to learn $\boldsymbol{\theta}$, we resort to the SEM formulation given in Equation (2), where we set $\mathbf{C}^0 = \mathbf{P}'\widehat{\mathbf{C}}^0\mathbf{P}$ (see Section 3.1). As a consequence, the causal tensor $\mathbf{C}^0$ in Figure 3 depends on $\prec$. According to the probabilistic model in Figure 3a, at each time-step we observe the vector $\mathbf{X}_T[t]$ by using a multivariate normal likelihood, precisely $MN(\mathbf{0}, \mathbf{MM}')$. Indeed, with constant causal coefficients and normally distributed noises, we have: (i) $\mathbb{E}[\mathbf{X}_T[t]] = \mathbf{M} \cdot \mathbb{E}[\mathbf{Z}[t]] = \mathbf{M} \cdot \mathbf{0} = \mathbf{0}$; (ii) $\mathrm{Var}[\mathbf{X}_T[t]] = \mathrm{Var}[\mathbf{MZ}] = \mathbf{M}\mathbb{I}\mathbf{M}' = \mathbf{MM}'$.

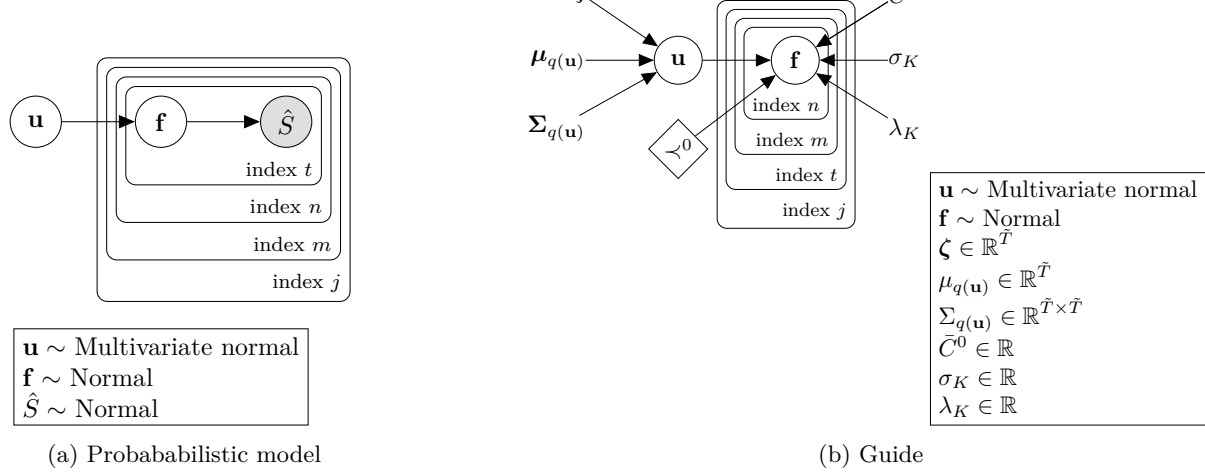

(a) Probababilistic model

(b) Guide

Figure 4: a Graphical models associated with the probabilistic model and b the parameterized variational distribution for learning batched GPs, along with variational parameters and their constraints.

**Model and guide for batched GPs inference.** Figures 4a and 4b depict the probabilistic model and the guide initialized at line 3 of Algorithm 1. Here we assume a known causal ordering, then we represent it with a rhomboid node. To impose such an ordering, we generate again a permutation tensor $\mathbf{P}$. To model each latent causal coefficient at a certain temporal resolution level $j$ as a smoothly varying function $\mathbf{f} \sim GP(\bar{C}_j^{(n,m)}, \mathbf{K}(t, t'))$, we exploit a variational formulation of Gaussian processes (Hensman et al., 2015). Accordingly, we consider a set of inducing points $\boldsymbol{\zeta} = \{\zeta_{\tilde{t}}\}_{\tilde{t}=1}^{\tilde{T}}$, optimised over the training set and where $\tilde{T} \leq T$ and latent inducing function variables $\mathbf{u}$ (a subset of $\mathbf{f}$) over these inducing points. Thus, the method relies upon the introduction of a joint variational distribution $q(\mathbf{f}, \mathbf{u})$, such that it factorises as $p(\mathbf{f} \mid \mathbf{u})q(\mathbf{u})$ (Titsias, 2009). This allows to avoid the computation of $\mathbf{K}_{\mathbf{ff}}^{-1}$ within the inference procedure. Here, to approximate the true GP prior $p(\mathbf{u})$ over the inducing points, we choose $q(\mathbf{u})$ to be a Cholesky variational distribution, i.e., a multivariate normal with positive definite covariance matrix $MN(\boldsymbol{\mu}_{q(u)}, \boldsymbol{\Sigma}_{q(u)})$. This variational approach allows to reduce the computational burden of GP estimation while avoiding overfitting at the same time (Bauer et al., 2016). In detail, in our work the usage of $\bar{T}$ inducing points lowers the computational cost of each GP from $\mathcal{O}(T^3)$ to $\mathcal{O}(\bar{T}^3)$ (Hensman et al., 2015). As a consequence, in both Figures 4a and 4b we have two latent variables: $\mathbf{u}$ associated with inducing functions and $\mathbf{f}$ associated with GP prior values. Here, we use batched GP to model causal coefficients, consequently they are independent both within and among time scales (rectangles in Figure 4). Since the joint distributions within the model and the guide $q(\mathbf{f}, \mathbf{u})$ factorise as;

$$p(\mathbf{f}, \mathbf{u}) = p(\mathbf{f} \mid \mathbf{u})p(\mathbf{u}); \qquad q(\mathbf{f}, \mathbf{u}) = p(\mathbf{f} \mid \mathbf{u})q(\mathbf{u}),$$

we also draw edges from $\mathbf{u}$ to $\mathbf{f}$. The variational parameters, along with their constraints, are shown in Figure 4b. In order to take into account and update only the causal relations that conform to the known ordering, we mask the distribution of the hidden functions above by using $\mathbf{P}$. Now, given these masked latent functions, we compute the mixing matrix $\mathbf{M}_j$. At this point, we observe the estimated $\widehat{\mathbf{S}}_j$ by using a Gaussian likelihood $N(\mathbf{M}_j\mathbf{M}'_j, \sigma)$, where the scale $\sigma \in \mathbb{R}_+$ is fixed (here we use .05). In particular, the mean value of the latter Gaussian is set in accordance with Section 3.2. To implement these probabilistic model and guide, we combine Pyro and GPyTorch (Gardner et al., 2018), an efficient Python library for GP inference built on PyTorch.

**SVI and stochastic optimizer.** Next, in line 4 we set the number of inference steps and choose the stochastic optimizer to be used within SVI. In our experiments, we use Adam (Kingma & Ba, 2014) along with learning rate decay and gradient clipping (Goodfellow et al., 2016). These tricks are useful to avoid bouncing around the optimal point when you are close to it and to prevent the gradient from becoming too large. Then, in line 5 and 6 we instantiate the SVI object for the inference of the causal ordering and for that of the batched GPs parameters, respectively. In order to initialize this objects, we specify the corresponding probabilistic model, guide, stochastic optimizer and the *ELBO* loss.

**Inference steps.** Subsequently, from line 7 to 10, we optimize the variational parameters in an alternating fashion. First, we optimize w.r.t. $\boldsymbol{\theta}$ to approximate the likelihood of $\prec$ given $\mathbf{X}_T[t]$. Unfortunately, the latent variable $\prec$ is non-reparameterizable. Then, in line 8 of Algorithm 1 we use the REINFORCE estimator (Williams, 1992), which is suitable for getting Monte-Carlo estimates of a certain cost function $f_{\boldsymbol{\phi}}(\mathbf{z})$. According to REINFORCE, we have

$$\nabla_{\boldsymbol{\phi}} \mathbb{E}_{q_{\boldsymbol{\phi}}(\mathbf{z})}[f_{\boldsymbol{\phi}}(\mathbf{z})] = \mathbb{E}_{q_{\boldsymbol{\phi}}(\mathbf{z})}[(\nabla_{\boldsymbol{\phi}} \log_{\boldsymbol{\phi}}(\mathbf{z}))f_{\boldsymbol{\phi}}(\mathbf{z}) + \nabla_{\boldsymbol{\phi}}f_{\boldsymbol{\phi}}(\mathbf{z})]. \tag{5}$$

Although unbiased, this estimator is known to have high variance. A way for reducing this variance is by means of control variate strategies, i.e., by adding a function within the expectation operator in Equation (5) that depends on the chosen values for $\mathbf{z}$ but is constant w.r.t. $\boldsymbol{\phi}$. So, the additional term does not affect the mean of the gradient estimator. Here, we resort to a data dependent baseline (Mnih & Gregor, 2014). The rationale behind the usage of baselines, is to reduce the variance by tracking the mean value of $f_{\boldsymbol{\phi}}(\mathbf{z})$. Thus, we add a running average of $f_{\boldsymbol{\phi}}(\mathbf{z})$, namely $\overline{f_{\boldsymbol{\phi}}(\mathbf{z})}$, for predicting the value of $f_{\boldsymbol{\phi}}(\mathbf{z})$ at each step. In our experiments, the usage of neural networks instead of running average did not lead to better results. After this inference step, we compute the mode of the $PL(\hat{\boldsymbol{\theta}})$, $\hat{\prec}^0$, in line 9 of Algorithm 1. Next, we optimize w.r.t. the rest of the variational parameters, by setting the causal ordering equal to $\hat{\prec}^0$. Since we isolate

the non-reparameterizable latent in the first step, in line 10 we exploit the reparameterization trick, to the benefit of learning. Finally, we return the learned variational parameters once the maximum number of iterations is reached.

## 4 Results

In this section we present the empirical assessment of our proposal. We first dive in the statistical analysis of the time series generated by the proposed model in Section 4.1. Then, Sections 4.2 and 4.3 present results regarding the inference of MN-DAGs from synthetic and real-world data, respectively.

### 4.1 Probabilistic Model over MN-DAGs

We start by illustrating the output of the proposed probabilistic generative model by means of an example. We consider $N = 3$ nodes (time series), $T = 512$ time steps, multiscale level $\mu = 0.5$, non-stationarity level $\tau = 0.5$, and density of causal interactions $\delta = 0.5$.

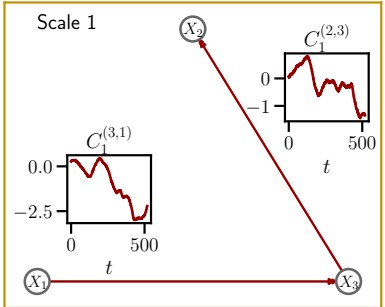 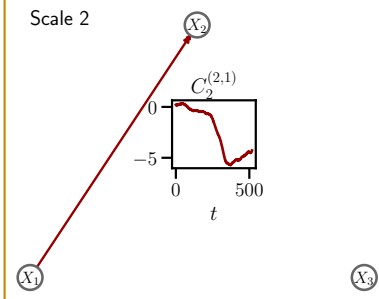 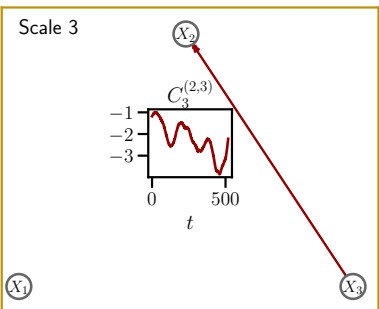

Figure 5: The figure depicts the latent MN-DAG sampled by using the proposed probabilistic generative model, where we set the number of nodes $N = 3$, number of time steps $T = 512$, multiscale level $\mu = 0.5$, non-stationarity level $\tau = 0.5$, and density parameter $\delta = 0.5$. The resulting MN-DAG has (i) 3 nodes; (ii) $J = 3$ pages (yellow rectangles), (iii) non-stationary causal interactions (red directed arrows, values shown as time series in the insets) that follow a Gaussian process with kernel $K = K_{\text{Periodic}} + K_{\text{Linear}} \times K_{\text{Matern}^{3/2}}$; (iv) global causal ordering $\prec = [1, 3, 2]$. Within each scale, we also plot the evolution of causal relations over time. Kernel variances are $\sigma_{\text{Linear}} = \sigma_{\text{Periodic}} = \sigma_{\text{Matern}^{3/2}} = 1$; the lengthscales $\lambda_{\text{Periodic}} = \lambda_{\text{Matern}^{3/2}} = 1/\tau$, and the period $\rho_{\text{Periodic}} = 1/\tau$. Given the kernel shape, the causal coefficients are locally periodic functions with increasing variation.

First, Figure 5 displays the underlying MN-DAG, sampled as detailed in Section 3.1, along with the evolution over time of causal relationships. We obtain an MN-DAG composed of three pages, corresponding to temporal resolutions $2^j$, $j = \{1, 2, 3\}$. The sampled causal ordering is $\prec = [1, 3, 2]$, and all causal relations, here locally periodic functions with increasing variations, are compliant with $\prec$. Indeed, we can only observe directed edges from time series $n$ to $m$, where $n \prec m$.

Now, given the sampled MN-DAG, we generate data according to Equation (4), where we use non-decimated Haar wavelet (Nason et al., 2000) as oscillatory function $\psi_j[t-k]$. Figure 6 depicts the generated time series along with descriptive statistics.

On the first row, we have the behaviour over time of synthetic data. Here, we resort to the augmented Dickey-Fuller test (ADF, Dickey & Fuller 1979) to assess stationarity. According to the test, the null hypothesis $H_0$ indicates that the process has a unit root (i.e., is non-stationary). The resulting $p$-values prove that the generated processes are (weakly) stationary (we reject $H_0$). Indeed, they have zero mean, while their dispersion looks different. On the one hand, the variance of $X_1$ (which occur at first position in $\prec$) is stationary, on the other hand those of $X_2$ and $X_3$ vary over time. Moreover, $X_2$, that has incoming causal edges at all temporal resolutions, displays the largest swings.

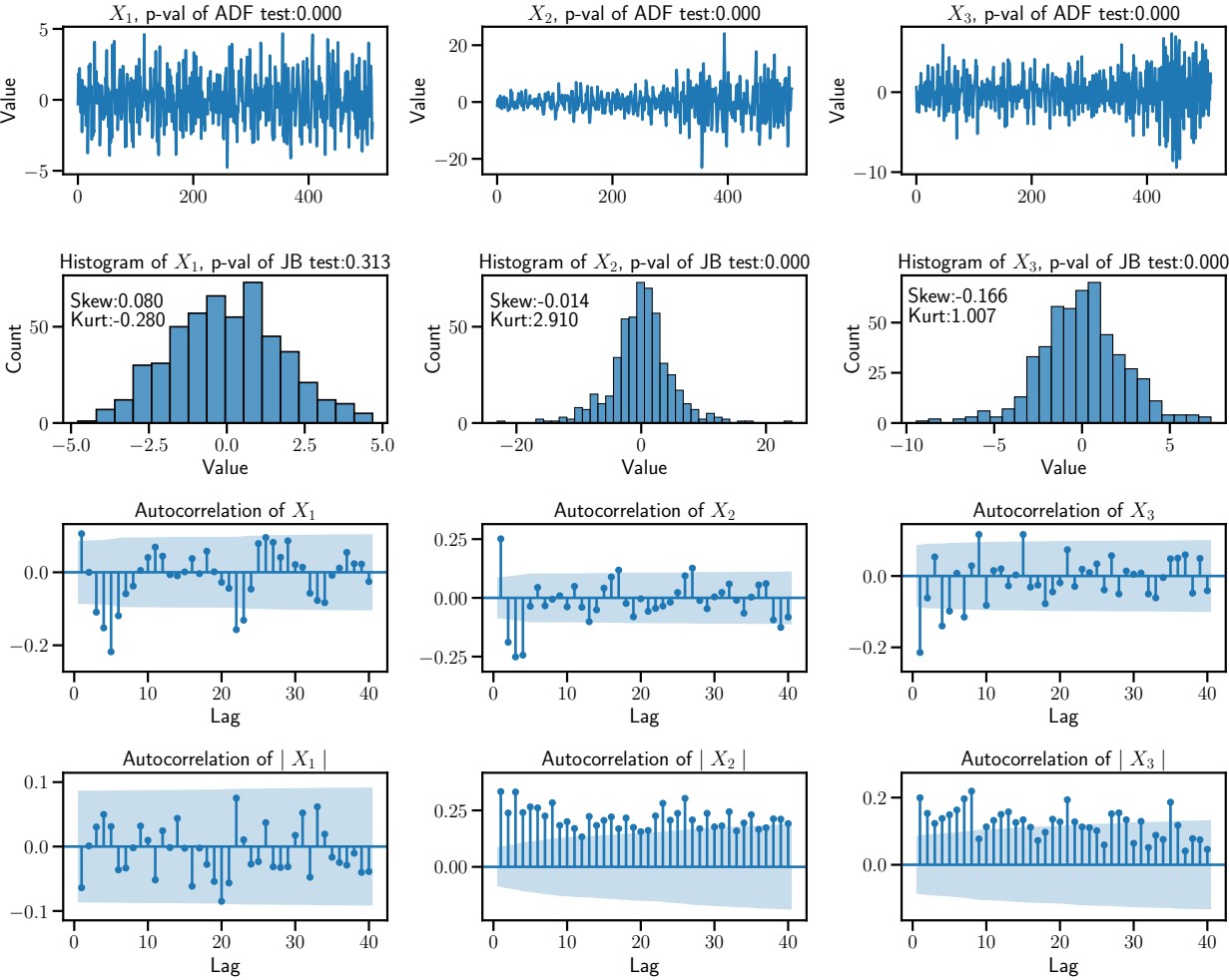

Figure 6: The figure shows the generated time series, along with descriptive statistics, where each process is associated with a different column. (i) Starting from the top, we have the synthetic data obeying to the underlying MN-DAG, where we provide the $p$-values of an ADF test. (ii) On the second row, we have the histograms of observed values, along with the $p$-values of a JB test, skewness, and kurtosis. (iii) The third row shows time series autocorrelations (with lag $l = \{1, \ldots, 40\}$). The light blue bands show 95% CIs. (iv) The last row shows the autocorrelations of absolute values of the processes.

On the second row we provide the histograms of the data, where we employ a Jarque-Bera test (JB, Jarque & Bera 1987) to assess normality. In particular, the null hypothesis $H_0$ is that the process is normally distributed. The resulting $p$-values suggest that $X_1$ is normally distributed, while we reject $H_0$ for both $X_2$ and $X_3$. Indeed, the associated distributions are leptokurtic, with $X_3$ having a more pronounced negative fat tail.

Looking at the autocorrelation (with lag $l \in [1, 40]$) plotted on the third row, we see that all the generated time series show serial correlation, statistically significant at 95% level (light blue bands). This result is in accordance with the multiscale nature of the time series. Moreover, the autocorrelation is driven by the local wavelet spectral matrix $\mathbf{S}_j$ (see Section 3.2), that in our model is determined by the causal structure.

Finally, the autocorrelation of absolute values of the processes prove that large swings in $X_2$ and $X_3$, either negative or positive, tend to be followed by other large swings. This effect is also known as volatility clustering, a key-feature of financial time series (Mandelbrot, 1967; Ding & Granger, 1996). Here, large movements in the series are driven by the increase of causal coefficients modulus, shown in Figure 5.

## 4.2 Causal Inference from Multiscale Data with Time-dependent Variance

We next report a comparison between our method, MN-CASTLE, the algorithm for multiscale causal structure learning introduced by D'Acunto et al. (2022), MSCASTLE, and state-of-the-art algorithms for learning Equation (2). For this comparison we use baselines belonging to different families, and synthetic data generated by the proposed probabilistic generative model. The goal is to assess the gain, in terms of performance, as we deviate from the single-scale stationary case, i.e., $\tau = \mu = 0$, which is the closest to Equation (2). Additionally, we report results concerning the inferred causal ordering and non-stationarity parameter $\hat{\tau}$.

**Settings.** We run our experiments according to three main different configurations. First, to evaluate the methods as we move within the $(\tau, \mu)$-quadrant, we generate the data by setting $N = 5$ and $T = 100$, while the entries of the $PL$ score vector are drawn from a uniform distribution $\theta_i \sim U(0, N)$. We test three values each for the multiscale and non-stationarity parameters, thus giving raise to configurations of none, medium, and high values for each parameter. For each possible combination $(\tau, \mu) \in \{0.0, 0.5, 0.9\} \times \{0.0, 0.5, 0.9\}$, we generate 20 datasets that contain $N$ time series each of length $T$. With regards the causal structure density, we use $\delta = 0.5$.

Second, to measure the sensitivity of the performances w.r.t. network density, we set $(N, T, \tau, \mu)$ equal to $(5, 100, 0.5, 0.5)$ and let $\delta$ varies in $\{0.25, 0.5, 0.75\}$. For each possible combination, we generate 20 datasets.

Third, to measure the sensitivity of the performances w.r.t. network size, we set $(T, \tau, \mu, \delta)$ equal to $(100, 0.5, 0.5, 0.25)$ and let $N$ varies in $\{5, 10, 15, 20\}$. Thus, in this experimental context we go from a configuration in which the number of observations $T$ is greater than the number of relationships possible in a complete single-scale DAG, i.e., $N \cdot (N - 1)/2$, to one in which it is much less. Also in this case, we generate 20 datasets for each combination.

Finally, in case $\tau \neq 0$, for the GP we use the radial basis function kernel $K_{\mathrm{RBF}}$ with variance $\sigma_{\mathrm{RBF}} = 0.1$ and lengthscale $\lambda_{\mathrm{RBF}} = 1/\tau$.

**Baselines.** We test MN-CASTLE against the following four baseline models. First we consider MSCAS-TLE, a multiscale causal structure learning model which exploits multiresolution analysis and non-convex continuous optimization to retrieve stationary causal relationships. Next, we have DirectLiNGAM (Shimizu et al., 2011), a method belonging to the family of non-Gaussian models. Algorithms within this class assume that the noise $\mathbf{Z}$ is non-normally distributed. Indeed, in this case the causal structure has shown to be fully identifiable (Shimizu et al., 2006a). Then, DirectLiNGAM returns an estimation of both causal ordering and causal coefficients. Second, we have CD-NOD (Huang et al., 2020), which belongs to the family of constraint-based methods. In particular, it has been developed to deal with heterogeneous (no assumptions on data distributions and causal relations) and non-stationary data as well. GOLEM (Ng et al., 2020) lives at the intersection of score-based and gradient-based methods. It solves an unconstrained optimization problem where the objective function is given by a likelihood function (as in score-based methods), penalized by regularization terms for sparsity and acyclicity.

As already mentioned, the concept of multiscale, non-stationary causal graphs is an understudied topic. Since none of the previous baseline models have been developed to infer causal graphs from data obeying to an underlying MN-DAG, our results provide information regarding the robustness of the previous algorithms with respect to the presence of multiple time scales and non-stationarity. In the following experiments, we use the code of MSCASTLE developed by D'Acunto et al. (2022); we exploit the implementations of DirectLiNGAM and GOLEM provided by `gCastle`[2] (Zhang et al., 2021), whereas we resort to `causallearn`[3] for the implementation of CD-NOD. The configuration for each baseline is provided in Appendix G.

Differently from MN-CASTLE, baseline models are non-probabilistic. While our model provides an approximate predictive posterior distribution over MN-DAGs, baseline models return a point estimate of an acyclic causal structure. In order to compare the algorithms, we retain all causal coefficients identified by MN-CASTLE that are in modulus significantly greater than 0.1 at 99% level. This post-processing implies that we can also obtain undirected edges for MN-CASTLE, since we do not impose any causal ordering.

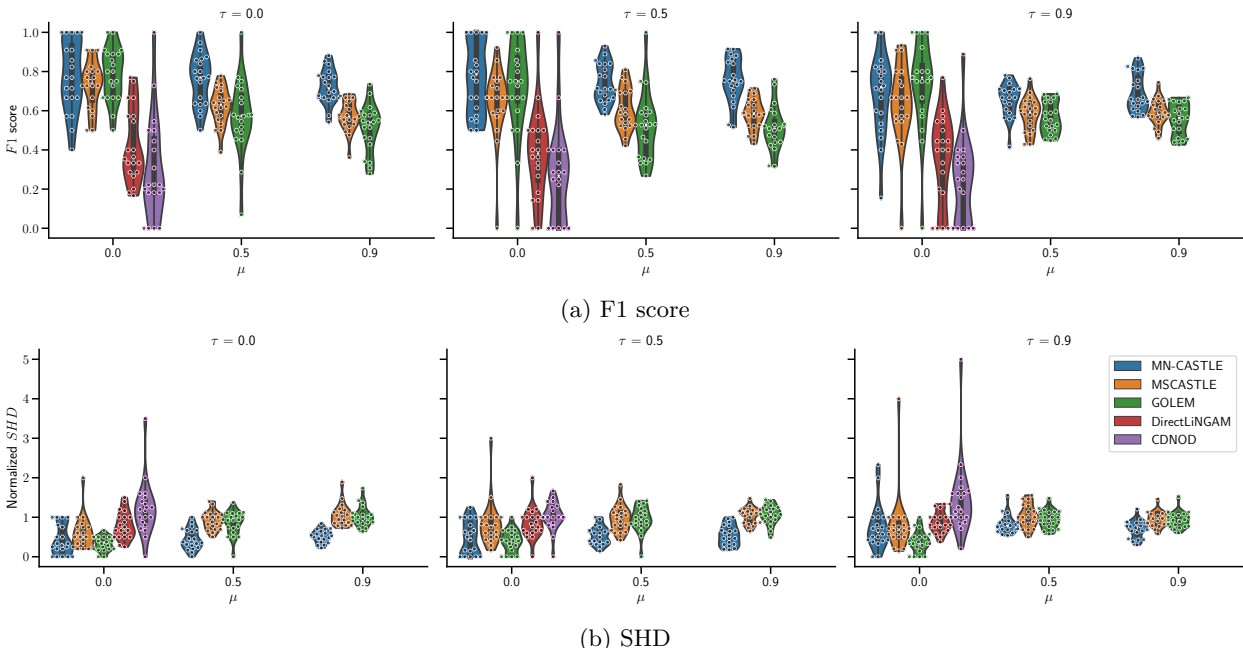

(a) F1 score

(b) SHD

Figure 7: The figure depicts the performances of the considered methods in the retrieval of the adjacency tensor, according to (a) F1 score and (b) structural Hamming distance (SHD). Higher F1 and lower SHD indicate better performance. Each model is associated to a different color. For every $(\tau, \mu)$ setting and every model, we represent the values attained over the 20 synthetic datasets through a violin plot. Within the latter, we provide a box-plot as well and we overlay the performances obtained for each dataset (points).

**Performance in the estimation of the adjacency tensor.** Retrieving the adjacency tensor means identifying the presence of causal relations disregarding their intensity. Appendix H provides an example of the evolving causal relations inferred by MN-CASTLE. Figures 7a and 7b refer to the first configuration described above, and show the F1 score (the higher the better) and *structural Hamming distance* (SHD, the lower the better) for the considered models, where SHD has been normalized by the number of edges present in the ground truth. The definition of the considered metrics are given in Appendix E. Appendix F also reports the values for additional evaluation scores and the fraction of undirected edges for MN-CASTLE.

Given a $(\tau, \mu)$ setting of the first configuration, for each model we have 20 values for every metric. The violin plot represents a kernel density estimation of the distribution of these 20 values, associated with a certain metric. Within each violin plot, we also report a box-plot in order to visualize the inter-quartile range (IQR). In addition, since the density is estimated by using only 20 points, we overlay the values of

---

[2]https://github.com/huawei-noah/trustworthyAI
[3]https://github.com/cmu-phil/causal-learn

the metrics attained for each of the 20 datasets, plotted as points. For each value of $\tau$, we provide the performances of each algorithm as $\mu$ varies. In case $\mu \neq 0$, we return the performance of GOLEM as well. In particular, because $\prec$ is global, we replicate the causal structure retrieved by the latter for each time scale. This way we obtain an additional baseline method also for multiscale datasets.

Overall, MN-CASTLE outperforms the baseline models in each case. When $\mu = 0$, the performances of MSCASTLE and GOLEM is very similar to that of our method. In contrast, as $\mu$ increases, we observe that the gap between MN-CASTLE and the other models widens and that, in general, the two multiscale models outperform GOLEM. Both F1 and SHD show that MN-CASTLE performance does not get lower as $\tau$ increases. In addition, when $\mu$ increases, the IQR tightens.

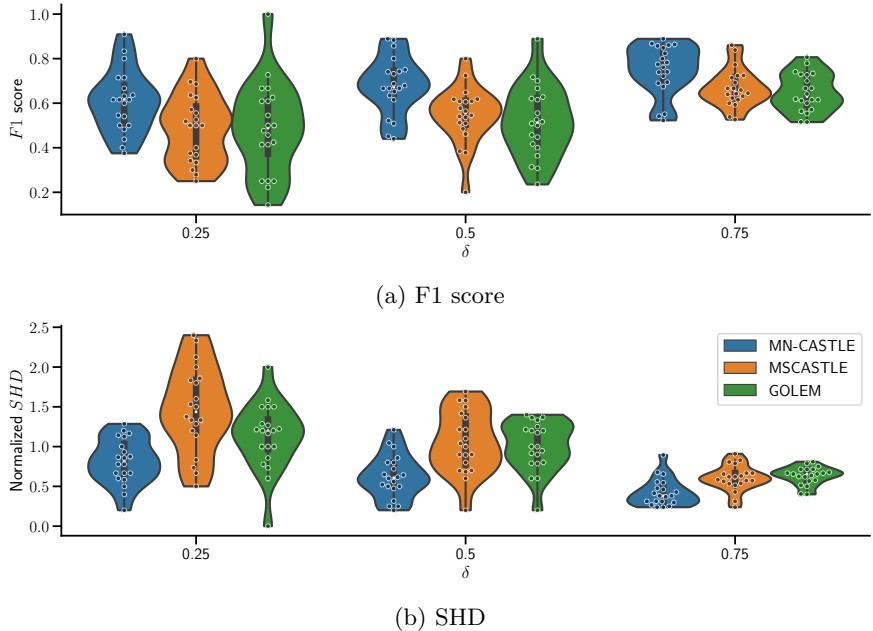

(a) F1 score

(b) SHD

Figure 8: The figure depicts the performances of the considered methods in the retrieval of the adjacency tensor, according to (a) F1 score and (b) structural Hamming distance (SHD). Higher F1 and lower SHD indicate better performance. Each model is associated to a different color. For every value of $\delta$ and every model, we represent the values attained over the 20 synthetic datasets through a violin plot. Within the latter, we provide a box-plot as well and we overlay the performances obtained for each dataset (points).

Figures 8a and 8b depict the performances of the models as we vary the density of the underlying MN-DAG. Since in this configuration, we use $\mu = 0.5$, we only retain MN-CASTLE, MSCASTLE and GOLEM. MN-CASTLE outperforms the baseline models in all settings, especially in terms of SHD. Moreover, the larger $\delta$, the better the performances of our model.

Figures 9a and 9b provide the performances of the methods as we vary the size of the underlying MN-DAG. Also in this setting, since we use $\mu = 0.5$, we only compare MN-CASTLE, MSCASTLE and GOLEM. In this case, the performances of the considered methods decrease as $N$ grows. Specifically, despite showing better metrics than baselines when N equals 5 and 10, MN-CASTLE underperforms MSCASTLE for N equals 15 and 20.

**Performance in the estimation of $\theta$ and $\tau$.** Figures 10a to 10c provide results concerning the goodness of the inferred vector of scores $\hat{\theta}$ of $PL$ distribution for the first experimental configuration, as measured by Spearman's rank correlation, normalized discounted cumulative gain (nDCG) at 3 and 5, w.r.t. the ground truth causal ordering $\prec$. Appendix E provides insights concerning the computation of these metrics, whereas Appendix F shows the results for Kendall-$\tau$ statistics. To represent the results, we use violin plots along with quartiles reference lines (dashed lines). For each of the 20 synthetic datasets, generated according to specific a pair $(\tau, \mu)$, we obtain an estimated $\hat{\theta}$. Then, we sample $10^3$ causal orderings $\hat{\prec}$ from $PL(\hat{\theta})$. Now,

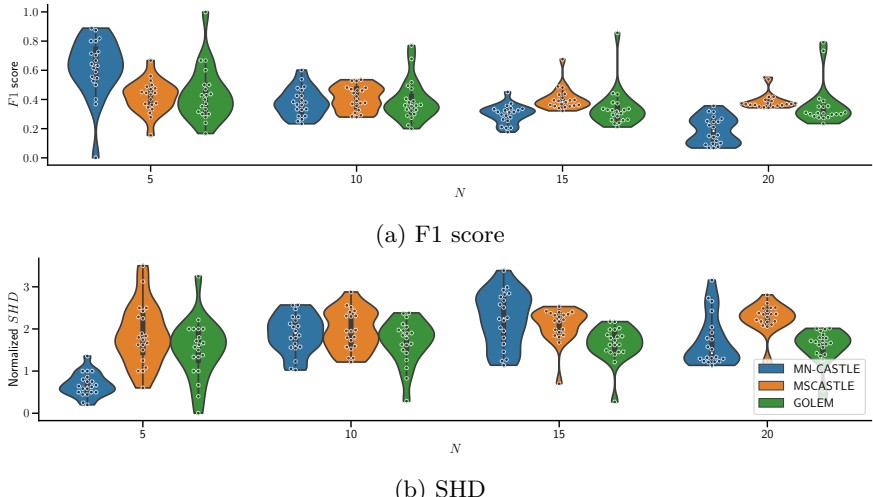

(a) F1 score

(b) SHD

Figure 9: The figure depicts the performances of the considered methods in the retrieval of the adjacency tensor, according to (a) F1 score and (b) structural Hamming distance (SHD). Higher F1 and lower SHD indicate better performance. Each model is associated to a different color. For every value of $N$ and every model, we represent the values attained over the 20 synthetic datasets through a violin plot. Within the latter, we provide a box-plot as well and we overlay the performances obtained for each dataset (points).

for each drawn causal ordering, we evaluate the three metrics w.r.t. $\prec$. As vector of scores for a baseline model, we use $\bar{\boldsymbol{\theta}}$, where $\bar{\theta}_i \sim U(0, N)$, $i = 1, \ldots, N$. Afterwards, we obtain $10^3$ random causal orderings $\bar{\prec}$ by sampling from the $PL$ distribution parameterized by $\bar{\boldsymbol{\theta}}$. As for MN-CASTLE, we evaluate the metrics w.r.t. $\prec$. Therefore, for each model, every violin plot is built by using $2 \times 10^4$ points. Overall, according to the monitored metrics, MN-CASTLE outperforms the baseline model. Moreover, the performances do not deteriorate as $\tau$ grows and improve as $\mu$ increases.

In addition, Figure 11 depicts the inferred values $\hat{\tau}$ for the non-stationarity parameter, obtained by means of the estimated GP kernel lengthscale, i.e., $\hat{\tau} = 1/\hat{\lambda}_{\mathrm{RBF}}$. Here red dashed lines refer to the ground truth value of $\tau$. In the stationary case, MN-CASTLE slightly overestimates the non-stationarity parameter for all values of $\mu$. When $\tau \neq 0$, on average our model correctly retrieve the ground truth for $\mu = .5$, whereas it slightly over/underestimates on average $\tau$ when $\mu$ is equal to .0 and .9, respectively.

We applied the same approach to evaluate the sensitivity of the estimation accuracy of $\theta$ and $\tau$ with respect to the density and number of nodes of the underlying MN-DAG. Figure 12 shows the results obtained on the synthetic data generated according to the second and third experimental settings, described above. Overall, the accuracy of MN-CASTLE in retrieving $\theta$ grows along with $\delta$: the IQR of the monitored metrics reach higher values and the spread of the estimated kernel densities reduces. Furthermore, unlike what we observe in metrics related to causal adjacency matrix estimation, the performance of MN-CASTLE in recovering causal ordering shows no dependence on $N$. Notice that here nDCG score depends on the value of $N$. In this way, we make the comparison meaningful by considering for each combination the same fraction of nodes.

Figure 13 provides the results related to estimation of $\tau$. Overal, MN-CASTLE tends to slightly overestimate $\tau$. However, no dependence of the model capability on $\delta$ and $N$ is observed.

## 4.3 Case Study on Global Equity Markets during the Covid-19 Pandemic

In this section, we apply MN-CASTLE to study the causal structure of 7 global stock markets: Hong Kong (HSI), Shanghai (SHC), Japan (NKX), United Kingdom (UKX), Germany (DAX), Italy (FMIB), US (SPX). Daily closing prices were collected from Stooq[4] and cover the period from January 2020 the 7[th] to December

---

[4]https://stooq.pl/

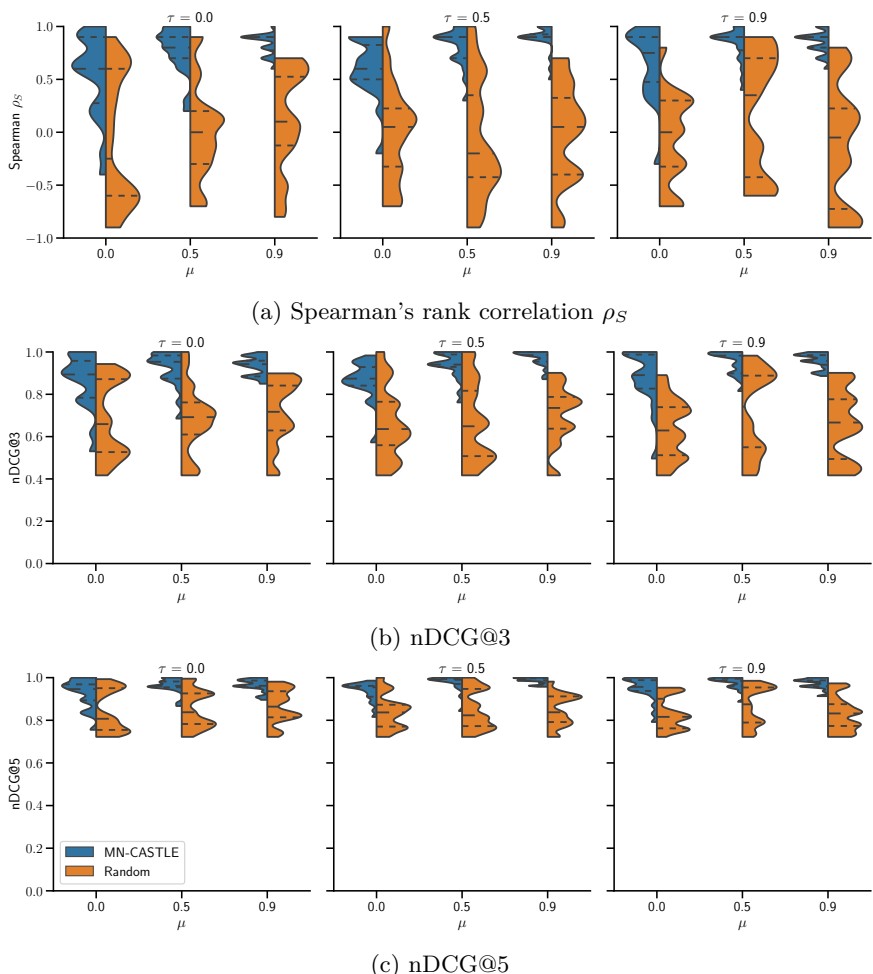

(a) Spearman's rank correlation $\rho_S$

(b) nDCG@3

(c) nDCG@5

Figure 10: The figure depicts violin plots along with quartiles reference lines (dashed lines) for (a) Spearman's rank correlation $\rho_S$, (b) normalized discounted cumulative gain (nDCG) at 3 and (c) 5. MN-CASTLE is given in blue while a random baseline model in orange. For every dataset generated according to a given $(\tau, \mu)$ setting (i) we sample $1 \times 10^3$ causal orderings $\hat{\prec} \sim PL(\hat{\boldsymbol{\theta}})$, where $\hat{\boldsymbol{\theta}}$ is the estimated vector of scores; (ii) we draw $1 \times 10^3$ random causal orderings $\bar{\prec} \sim PL(\bar{\boldsymbol{\theta}})$, where $\bar{\theta}_i \sim U(0, N)$, $i = 1, \ldots, N$. Afterwards, we evaluate the three metrics by using the sampled causal orderings and $\prec$ for both models. Thus, each violin plot is made by $2 \times 10^4$ points.

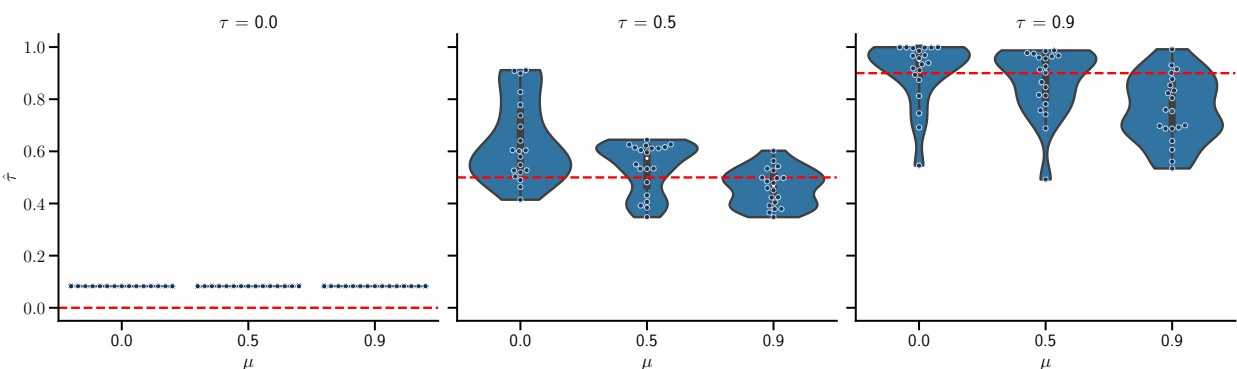

Figure 11: The figure illustrates the violin plots concerning the estimated values $\hat{\tau}$ for the non-stationarity parameter. Red dashed lines refer to the ground truth value of $\tau$.

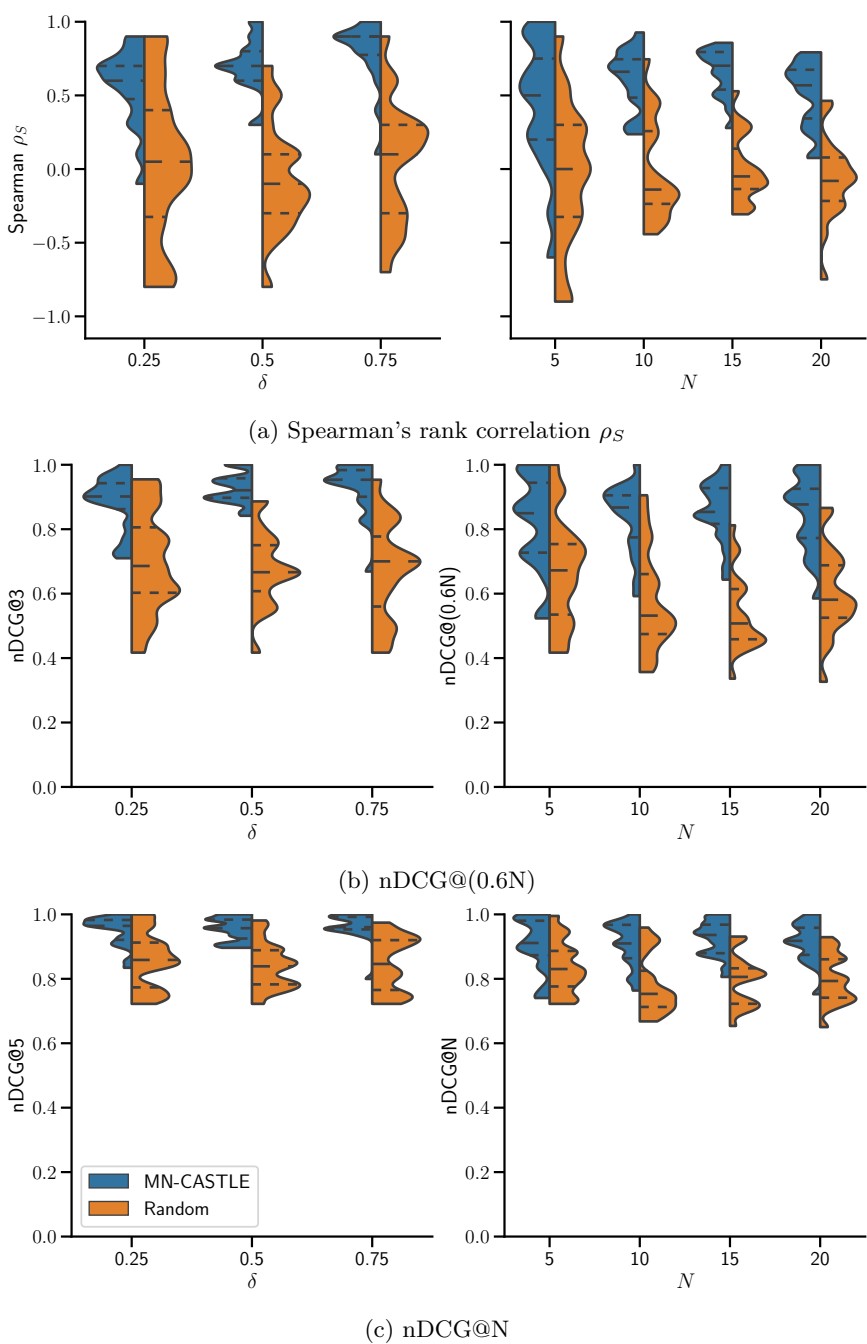

(a) Spearman's rank correlation $\rho_S$

(b) nDCG@(0.6N)

(c) nDCG@N

Figure 12: The figure depicts violin plots along with quartiles reference lines (dashed lines) for (a) Spearman's rank correlation $\rho_S$, (b) normalized discounted cumulative gain (nDCG) at $0.6 \cdot N$ and (c) $N$. MN-CASTLE is given in blue while a random baseline model in orange. Violin plots on the left refers to the second experimental configuration, i.e., when we vary $\delta$ while keeping fixed the values of the others parameters, as described above. Violin plots on the right concerns the third experimental setting, where we study the sensitivity of the estimation accuracy w.r.t. the network size $N$.

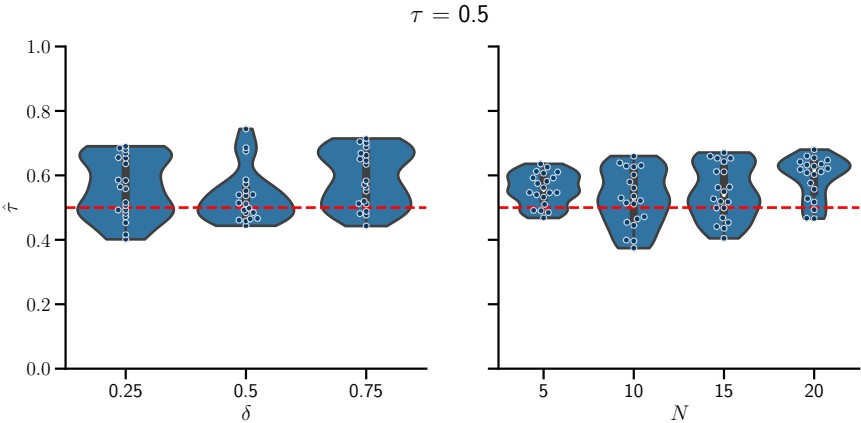

Figure 13: The figure illustrates the violin plots concerning the estimated values $\hat{\tau}$ for the non-stationarity parameter. Red dashed lines refer to the ground truth value of $\tau$. On the left, we have the results for the second experimental setting, and on the right are those for the third.

2021 the 31[st]. From the values of the above indices, we calculate logarithmic returns[5], which are used to infer the causal ordering and estimate the power spectrum.

In order to obtain an estimate of the latter at 95% confidence level, we used the R package mvLSW (Taylor et al., 2019). The wavelet transform is performed using Daubechies wavelet with filter length equal to 8. Moreover, the smoothing of the periodogram is performed using the rectangular kernel, also known Daniell window, of width 22. Finally the smoothed periodogram is corrected for the bias by using the inverted autocorrelation wavelet (Eckley & Nason, 2005) and regularized to ensure positive definiteness (Schnabel & Eskow, 1999).

We obtain cross-spectra significantly different from zero only for the scale level $j = 1$ and over the first semester of 2020. Accordingly, we restrict our analysis to the period from January 2020 the 7[th] to June 2020 the 30[th] and feed MN-CASTLE with the power spectrum values for the finest scale level. To infer the causal coefficients we use the radial basis function kernel.

Figure 14a shows the estimated causal graph for $j = 1$, where node are coloured according to the geographical area. In addition, we report the retrieved causal ordering $\hat{\prec}^0$ and the vector $\theta$ containing the score associated to each stock market, as well. We obtain that, over the first semester of 2020, returns of Asian equity markets causally impact those of both European and American ones. In particular, Hong Kong and Shanghai occupy the first and second positions in the causal ordering, respectively. Next we find the European stock markets, which in turn causally influence the US stock market. From the value $\theta$, we see that MN-CASTLE assigns almost the same score to UKX and NKX.

Looking at the behaviour of causal coefficients along time given in Figure 14b, we see that causal relations among markets are (i) time-dependent, where the estimated non-stationarity parameter is $\hat{\tau} = 0.80$; (ii) mainly positive; (iii) tend to grow in the first half of the semester, after which they decrease. In addition, the sign of the causal coefficient changes for several relationships and some (e.g., HSI→SHC, SHC→UKX) become statistically nondifferent from zero in the second half of the analyzed period. We also notice a statistically significant connection from NKX→UKX, that is not shown in Figure 14a since not compliant to the estimated mean causal ordering $\prec^0$. The presence of such a connection underscores the uncertainty of MN-CASTLE regarding the positioning of UKX and NKX within the causal order.

Finally, we compare MN-CASTLE estimated network with those retrieved by MSCASTLE and GOLEM over the same time period, given in Figures 15a and 15b. In case of MSCASTLE, we use the same wavelet transform as above, i.e., Daubechies wavelet with filter length equal to 8, and we consider only the statistically significant scale level $j = 1$.

---

[5]Given the prices of the $i$-th stock market index at the time $t$ and $t-1$, namely $P_t$ and $P_{t-1}$, the logarithmic return at time $t$ is given by $r_{i,t} = \ln (P_t/P_{t-1})$.

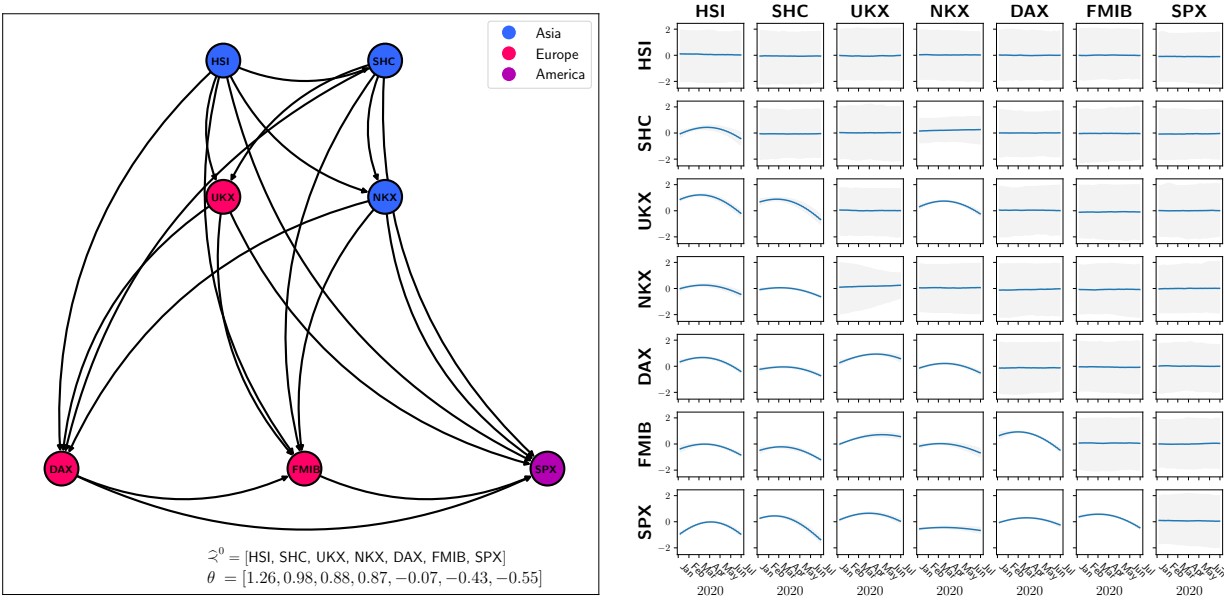

(a) Inferred causal structure, mean causal ordering $\hat{\prec}^0$ and PL parameters vector $\theta$.

(b) Causal coefficients. Financial markets are sorted according to the mean causal ordering $\hat{\prec}^0$.

Figure 14: The figure depicts a the inferred causal structure and b estimated causal coefficients along with 95% CI (shaded bands) over the first semester of 2020, for the finest scale level $j = 1$.

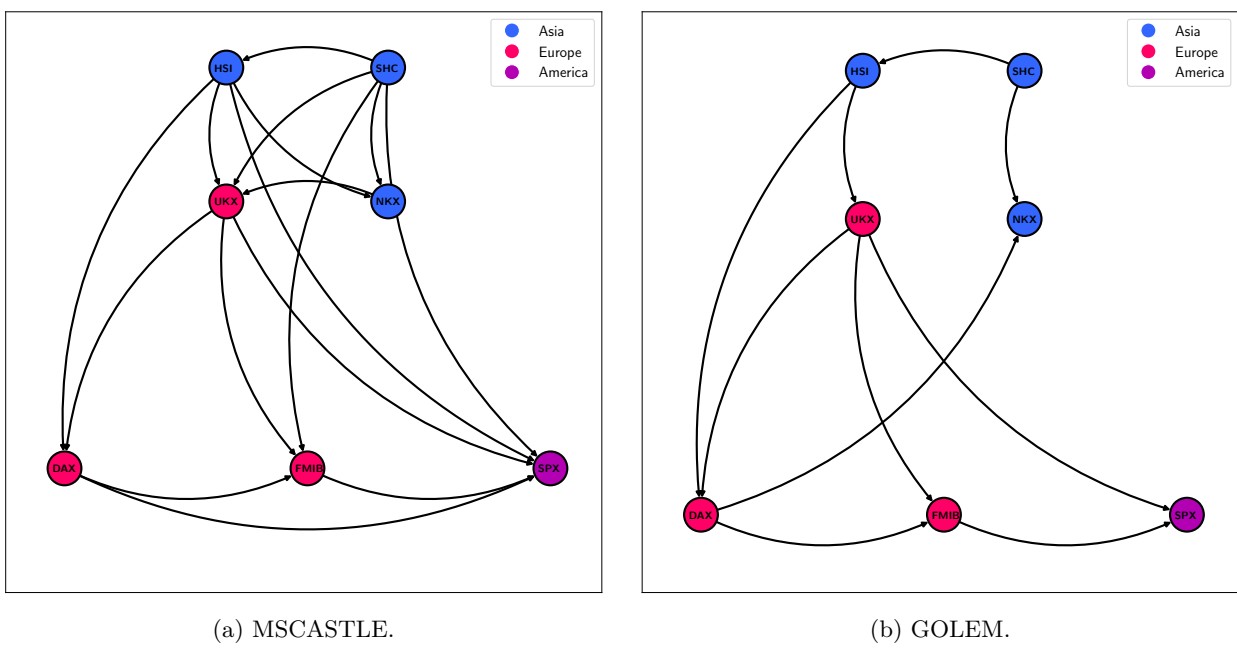

(a) MSCASTLE.

(b) GOLEM.

Figure 15: The figure depicts the causal structures inferred by a MSCASTLE and b GOLEM, over the first semester of 2020. To easy the comparison, we position the nodes according to the causal ordering estimated by MN-CASTLE.

Overall, multiscale models estimate denser networks than that inferred by GOLEM. To quantify the similarity between the causal adjacency matrices estimated by the models, regardless of the causal relation sign, we

Table 1: Jaccard score between estimated causal adjacency matrices.

|  | MSCASTLE | GOLEM |
|---|---|---|
| MN-CASTLE | 0.64 | 0.36 |
| MSCASTLE |  | 0.53 |

estimate the Jaccard score[6], reported in Table 1. The greatest agreement occurs between the networks returned by MN-CASTLE and MSCASTLE.

## 5 Conclusions and Future Research Directions

This paper deals with multiscale non-stationary causal analysis, filling a gap in the literature. Indeed, the bulk of previous work assume that the only relevant temporal resolution for causal relations is the frequency of observed data. We drop such assumption. Moreover, we also allow the causal relations to vary over time. Since in general there is no prior knowledge about the relevant time scales of causal interactions nor about their temporal dependencies, the proposed framework of MN-DAGs represents an important step in such direction.

**Generative model.** We propose a novel probabilistic model to generate time series data obeying to an underlying MN-DAG, in accordance with the specified values for multiscale and non-stationary features, $\mu$ and $\tau$, respectively. Our model leverages the well established mathematical theory of multivariate locally stationary wavelet processes and linear structural equation model. The causal ordering is modeled by means of Plackett-Luce distribution while the causal interactions evolve over time according to the specified kernel of a Gaussian process. Statistical analysis of generated data proves the exposed model to be able to reproduce well-known features of time series. Therefore, it represents a suitable framework for testing the robustness of causal inference methodologies on datasets generated from different points of the $(\tau, \mu)$-quadrant shown in Figure 1.

We stress the importance of providing both researchers and practitioners with synthetic data generators capable of replicating phenomena characterizing data from different application domains.

Future work should aim to overcome some limitations related to the framework adopted to manage different time-resolutions and the modeling of the causal tensor. In particular, multivariate locally stationary wavelet processes formulation relies upon wavelets, that are known to suffer from limited joint time-frequency resolution (Heisenberg uncertainty principle). Indeed, wavelets divide the frequency space into non-overlapping bands, i.e., octave bands. Furthermore, since the auto/cross-correlation structure of generated data depends of both the power spectrum decomposition across temporal scales and the auto-correlation wavelet, the usage of diverse wavelet families might lead to different results. Then, the usage of alternative methods to wavelet transform might improve the proposed generative model. With regards the causal tensor, structural breaks such as sudden deletion/addition of causal edges might be added within Equation (1).

From a theoretical point of view, an interesting research direction is to study the assumptions that make the model described in Eq. (4) identifiable. Even though some class of linear structural equation models have been proved identifiable under different types of restrictions (Shimizu et al., 2006b; Peters & Bühlmann, 2014; Loh & Bühlmann, 2014; Park & Kim, 2020), the case of MN-DAG needs to be carefully investigated. Indeed, the presence of the non-decimated wavelet transform; the unobservability of the contributions to the process coming from each time resolution; the linearity of the model in the frequency domain are some of the points that distinguish the MN-DAG case from those currently studied.

**Bayesian causal inference method.** In addition, we expose a Bayesian method for learning MN-DAGs from time series data, termed MN-CASTLE. The latter relies upon observed time series data and an estimate for the power spectrum at each scale level. We implement the latter by using an alternating two-steps approach. In the first step we optimize w.r.t. the Plackett-Luce vector of scores $\boldsymbol{\theta}$, by using the values of

---

[6]Given two edge sets $\mathcal{E}_1$ and $\mathcal{E}_2$, this score is defined as $Jacc(\mathcal{E}_1, \mathcal{E}_2) = |\mathcal{E}_1 \cap \mathcal{E}_2|/|\mathcal{E}_1 \cup \mathcal{E}_2|$. Moreover, $Jacc(\mathcal{E}_1, \mathcal{E}_2) \in [0, 1]$.

time series at time $t$. Then, we keep the causal ordering fixed to the mode of the Plackett-Luce distribution, i.e., $\hat{\prec}^0$, and we make a stochastic variational inference step on the rest of variational parameters related to the causal coefficient tensor by exploiting the provided estimation for the power spectrum. Even though this procedure has no theoretical guarantee to converge towards the global optimum, we observe a smoother learning process for the weights of causal edges.

Our findings show that MN-CASTLE compares favorably to baseline models in the retrieval of the adjacency tensor of the causal graph. We test the models on synthetic datasets generated according to different $(\tau, \mu)$ configurations, from the single-scale stationary to highly multiscale non-stationary case. We observe that the performance of MN-CASTLE, depicted in Figure 7, is not sensitive to the value of non-stationarity parameter. On the contrary, the growth of the multiscale parameter is associated with an improvement in the quality of the results returned by our method. This trend is also shown in Figure 10, that concerns the goodness of the estimated vector of scores for Plackett-Luce distribution. On one hand, we think that when non-stationarity and multiscale parameters are different from zero, MN-CASTLE might benefits of greater differences among time series distributions. On the other hand, we believe that the large variance shown (especially in the single scale case) is an effect due to the low cardinality of the edge set. In fact, even though the monitored metrics are normalized, on average in the single scale case we only have five causal links. So, a single error weighs more. In addition, MN-CASTLE correctly tracks the underlying value of the non-stationarity parameter, which means that is able to provide accurate information related to the evolution of causal relations (see Appendix H). We emphasize that MN-CASTLE, being a fully Bayesian approach, by definition takes into account uncertainty. Consequently, we sample MN-DAGs from the approximate posterior distribution in accordance with the confidence of the model.

Furthermore, we also study the behaviour of our model w.r.t. the density $\delta$ of the underlying MN-DAG. We observe that the performance of MN-CASTLE improves as $\delta$ increases and that our method outperforms the other models in all cases. This improvement is also manifest in the value of the metrics used to evaluate the estimated causal ordering. We also provide supplementary results on additional synthetic data to test the capabilities of the monitored methods when the MN-DAG size $N$ increases, keeping the other parameters fixed. We note that the performance of the models in retrieving the causal adjacency matrix worsens as $N$ increases. This is somewhat expected, as we move from a configuration in which the number of observations $T$ is greater than the number of possible causal connections to be estimated, to one in which it is much less. Although MN-CASTLE continues to outperform the baseline models in the case where $N$ is 5 or 10, we observe that its performance deteriorates more rapidly than that returned by the baselines. In contrast to what we observe about the causal adjacency matrix, the ability of MN-CASTLE in estimating causal ordering does not deteriorate as N increases. Hence, we think that the impairment of our model shown in Figure 9 arises from the estimation of the batched GPs.

As a case study, we apply MN-CASTLE to retrieve the causal structure underlying the returns of 7 global equity markets, during Covid-19 pandemic. Since our method relies upon multi-resolution analysis, it looks for causal links only at relevant temporal resolutions. From the analysis of the power spectrum, we obtain that important causal relationships occur at the finest scale level, which corresponds to 2-4 days time resolution. We believe this is justified by the fact that the period analyzed is characterized by financial turbulence, due to the outbreak of the pandemic. During this crisis, investors had to react quickly to the shocks that followed, generating sudden swings in stock prices.

Our findings show that Asian markets are the main drivers of returns within the considered network. Next, we find European stock markets, that influence the US one. Therefore, our results show that causal relationships tend to reflect the spread of the epidemic. Indeed, according to the World Health Organization (WHO) situation reports, in the early stage of the pandemic, we observe a higher number of confirmed cases in Europe than in United States (WHO, 2020-03-15). Although this may not reflect the actual spread of the virus, it does indicate that a more effective level of surveillance was implemented in European countries early on. Moreover, the positivity of the causal relationships and the fact that they grow in magnitude during the first months of 2020 are consistent with the outbreak of the epidemic. This, together with the statistical significance of the cross-spectrum over the first half of 2020, confirms that the estimated causal interactions are not the result of time lag in stock market closures.

The network inferred by our method is more similar to that of MSCASTLE than that estimated by GOLEM, as indicated by the Jaccard score values in Table 1. Due to their multiscale analysis, MN-CASTLE and MSCASTLE are able to identify more connections between the stock markets considered. In addition, MN-CASTLE also provides information on the evolution of these connections, highlighting crucial aspects such as statistical significance, weakening/strengthening and any sign changes over the time period analysed. These issues, may impair baseline methods, which assume causal relationships to be stationary, in recovering some connections. Finally, Figures 15a and 15b also demonstrate that the mean causal ordering estimated by our method is respected by the baselines, in most cases.

Future research directions to improve MN-CASTLE concern (i) the definition of a single-step inference procedure able to limit the variance of the gradient estimator due to the presence of Plackett-Luce distribution; (ii) overcoming the limitations highlighted by Figure 9; (iii) the inference of nonlinear causal relations; (iv) modeling of causal ordering that may vary on different time scales; (v) relaxation of the assumption concerning the presence of a single kernel shared by all causal relationships. The latter point is important from an application point of view, since the order of magnitude of power spectrum values can vary greatly, especially between different time resolutions. However, since this would mean increasing the number of model parameters considerably, performance could be negatively affected.

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

## Appendix A   Overview of SVI

*Stochastic variational inference* (Hoffman et al., 2013; Kingma & Welling, 2013) is an algorithm that combines *variational inference* (VI, Blei et al. 2017) and *stochastic optimization* (Spall, 2005). SVI approximates the posterior distribution of complex probabilistic models that involves hidden variables, and can handle large datasets. Consider a dataset $\mathcal{X} = \{\mathbf{x}^{(i)}\}_{i=1}^{T}$ of $T$ i.i.d. samples of either a continuous or discrete variable $\mathbf{x}$. Suppose that $\mathcal{X}$ is generated according to a latent continuous random variable $\mathbf{z}$, The latter is governed by a vector of parameters $\boldsymbol{\beta}^*$ endowed with a prior distribution $p(\boldsymbol{\beta}^*)$ , i.e., $\mathbf{z}^{(i)} \sim p_{\boldsymbol{\beta}^*}(\mathbf{z})$. Thus, we have data are generated according to a conditional distribution, i.e., $\mathbf{x}^{(i)} \sim p_{\boldsymbol{\beta}^*}(\mathbf{x} \mid \mathbf{z})$. Both the prior $p_{\boldsymbol{\beta}^*}(\mathbf{z})$ and the conditional distribution $p_{\boldsymbol{\beta}^*}(\mathbf{x} \mid \mathbf{z})$ belong to parametric families of distributions $p_{\boldsymbol{\beta}}(\mathbf{z})$ and $p_{\boldsymbol{\beta}}(\mathbf{x} \mid \mathbf{z})$ whose PDFs are differentiable w.r.t. $\boldsymbol{\beta}$ and $\mathbf{z}$. Our goal is to compute the likelihood of the hidden variable given the observations, i.e., the posterior

$$p_{\boldsymbol{\beta}}(\mathbf{z} \mid \mathbf{x}) = \frac{p_{\boldsymbol{\beta}}(\mathbf{x}, \mathbf{z})}{\int p_{\boldsymbol{\beta}}(\mathbf{x}, \mathbf{z}) \, d\mathbf{z}} \,. \tag{6}$$

Since the denominator of Equation (6), also known as evidence, is usually intractable to compute, a well-known solution is to approximate the target posterior. Within approximate posterior inference methodologies, VI casts learning as an optimization problem. More in details, VI involves the introduction of a family of variational distributions $q_{\boldsymbol{\phi}}(\mathbf{z} \mid \mathbf{x})$, parameterized by a variational parameters $\boldsymbol{\phi}$. Then, VI optimzes those parameters to find $q_{\boldsymbol{\phi}^*}(\mathbf{z} \mid \mathbf{x})$, i.e., the member of the variational distributions family that is closest to the posterior distribution. Here closeness is measured according to Kullback-Leibler divergence (KL).

The objective of SVI is the *evidence lower bound* (*ELBO*), that is equal to the negative KL divergence up to a term that does not depend on $q$

$$\begin{aligned} ELBO &= \mathbb{E}_{q_{\boldsymbol{\phi}}(\mathbf{z}|\mathbf{x})}[\log p_{\boldsymbol{\beta}}(\mathbf{x}, \mathbf{z}) - \log q_{\boldsymbol{\phi}}(\mathbf{z} \mid \mathbf{x})] \\ &= -D_{KL}(q_{\boldsymbol{\phi}}(\mathbf{z} \mid \mathbf{x}^{(i)}) \| p_{\boldsymbol{\beta}}(\mathbf{z} \mid \mathbf{x}^{(i)})) + \log p_{\boldsymbol{\beta}}(\mathbf{x}^{(i)}) \\ &= -D_{KL}(q_{\boldsymbol{\phi}}(\mathbf{z} \mid \mathbf{x}^{(i)}) \| p_{\boldsymbol{\beta}}(\mathbf{z})) + \mathbb{E}_{q_{\boldsymbol{\phi}}(\mathbf{z}|\mathbf{x}^{(i)})}[\log p_{\boldsymbol{\beta}}(\mathbf{x}^{(i)} \mid \mathbf{z})] \,. \end{aligned} \tag{7}$$

Since KL is a non-negative measure of closeness between distributions, then $\log p_{\boldsymbol{\beta}}(\mathbf{x}) \geq ELBO$ for all $\boldsymbol{\beta}$ and $\boldsymbol{\phi}$. Therefore, the maximization of the *ELBO* is equivalent to the minimization of the distance between $q_{\boldsymbol{\phi}}(\mathbf{z})$ and $p_{\boldsymbol{\beta}}(\mathbf{x} \mid \mathbf{z})$. Observations $\mathbf{x}^{(i)}$ are conditionally independent given the latent, thus the log likelihood term in Equation (7) can be written as

$$\sum_{i=1}^{T} \log p(\mathbf{x}^{(i)} \mid \mathbf{z}) \approx \frac{T}{T'} \sum_{i \in \mathcal{I}_{T'}} \log p(\mathbf{x}^{(i)} \mid \mathbf{z}) \,,$$

where $\mathcal{I}_{T'}$ is a set of indexes of size $T' \leq T$. One way to subsample indexes is, for example, to randomly select $T'$ data points among the observations Thus, in case of large datasets, we can run SVI while exploiting mini-batch optimization.

In order to compute the gradient of the *ELBO* w.r.t. $\boldsymbol{\phi}$, SVI relies upon the reparameterization trick. The continuous random variable $\mathbf{z}$ can be expressed in terms of a deterministic function $\mathbf{z} = g_{\boldsymbol{\phi}}(\boldsymbol{\epsilon}, \mathbf{x})$, where $\boldsymbol{\epsilon} \sim q(\boldsymbol{\epsilon})$ is independent of $\mathbf{z}$. This procedure is useful to move all the dependence on $\boldsymbol{\phi}$ inside the expectation operator

$$\mathbb{E}_{q_{\boldsymbol{\phi}}(\mathbf{z}|\mathbf{x}^{(i)})}[f_{\boldsymbol{\phi}}(\mathbf{z})] = \mathbb{E}_{q(\boldsymbol{\epsilon})}[f_{\boldsymbol{\phi}}(g_{\boldsymbol{\phi}}(\boldsymbol{\epsilon}, \mathbf{x}^{(i)}))] \,,$$

where $f_{\boldsymbol{\phi}(\mathbf{z})}$ represents a general cost function. Now, the gradient can be computed as

$$\begin{aligned} \nabla_{\boldsymbol{\phi}} \mathbb{E}_{q(\boldsymbol{\epsilon})}[f_{\boldsymbol{\phi}}(g_{\boldsymbol{\phi}}(\boldsymbol{\epsilon}, \mathbf{x}^{(i)}))] &= \mathbb{E}_{q(\boldsymbol{\epsilon})}[\nabla_{\boldsymbol{\phi}} f_{\boldsymbol{\phi}}(g_{\boldsymbol{\phi}}(\boldsymbol{\epsilon}, \mathbf{x}^{(i)}))] \\ &\approx \frac{1}{L} \sum_{l=1}^{L} f(g_{\boldsymbol{\phi}}(\boldsymbol{\epsilon}^{(i,l)}, \mathbf{x}^{(i)})) \,, \end{aligned}$$

where $L$ is the number of samples per data point. Then, we obtain an unbiased estimate of the gradient by means of Monte-Carlo estimates of this expectation.

## Appendix B  Existence of the Inverse

To prove invertibility of $\mathbb{I} - \mathbf{C}$, $\mathbf{C} \in \mathbb{R}^{N \times N}$, let us rewrite $\mathbf{C} = \mathbf{P}'\widetilde{\mathbf{C}}\mathbf{P}$. Here, $\mathbf{P} \in \mathbb{R}^{N \times N}$ is a permutation matrix entailed by the causal ordering $\prec$, such that $p_{nn'} = 1$ iff the node $X_n$ occurs at position $n'$ within $\prec$, and $\widetilde{\mathbf{C}}$ is a strictly lower triangular matrix, computed by ordering the rows of $\mathbf{C}$ according to $\prec$. Now, for permutation matrices it holds $\mathbf{P}^{-1} = \mathbf{P}'$. In addition, since strictly lower triangular matrices are nilpotent, there exists an integer $\bar{N}$ such that $\widetilde{\mathbf{C}}^{\bar{n}} = 0$, $\forall \bar{n} \geq \bar{N}$. Then it follows that $\mathbf{C}$ is similar to $\widetilde{\mathbf{C}}$ and, consequently, nilpotent too:

$$
\begin{aligned}
\mathbf{C}^{\bar{N}} &= (\mathbf{P}'\widetilde{\mathbf{C}}\mathbf{P})^{\bar{N}} \\
&= (\mathbf{P}^{-1}\widetilde{\mathbf{C}}\mathbf{P})^{\bar{N}} \\
&= (\mathbf{P}^{-1}\widetilde{\mathbf{C}}\mathbf{P})(\mathbf{P}^{-1}\widetilde{\mathbf{C}}\mathbf{P})\dots(\mathbf{P}^{-1}\widetilde{\mathbf{C}}\mathbf{P}) \\
&= \mathbf{P}^{-1}\widetilde{\mathbf{C}}(\mathbf{P}\mathbf{P}^{-1})\widetilde{\mathbf{C}}(\mathbf{P}\mathbf{P}^{-1})\dots(\mathbf{P}\mathbf{P}^{-1})\widetilde{\mathbf{C}}\mathbf{P} \\
&= \mathbf{P}^{-1}\widetilde{\mathbf{C}}^{\bar{N}}\mathbf{P} \\
&= 0 \,.
\end{aligned}
$$

At this point, exploiting the geometric series representation (nilpotent matrices have eigenvalues equal to zero and then are convergent), we have that

$$
\begin{aligned}
(\mathbb{I} - \mathbf{C})^{-1} &= \sum_{\bar{n}=0}^{\infty} \mathbf{C}^{\bar{n}} \\
&= \sum_{\bar{n}=0}^{\bar{N}-1} \mathbf{C}^{\bar{n}} \,.
\end{aligned}
$$

Therefore, the inverse exists and is given by a finite sum of powers of $\mathbf{C}$.

## Appendix C  M is a Permuted Lower Triangular Matrix

Starting from the representation of $\mathbf{C} = \mathbf{P}'\widetilde{\mathbf{C}}\mathbf{P}$ as in Appendix B, we have:

$$
\begin{aligned}
\mathbf{M} &= (\mathbf{I} - \mathbf{P}'\widetilde{\mathbf{C}}\mathbf{P})^{-1} \\
&= (\mathbf{P}^{-1}\mathbf{P} - \mathbf{P}'\widetilde{\mathbf{C}}\mathbf{P})^{-1} \\
&= (\mathbf{P}'\mathbf{P} - \mathbf{P}'\widetilde{\mathbf{C}}\mathbf{P})^{-1} \\
&= (\mathbf{P}'(\mathbf{I} - \widetilde{\mathbf{C}})\mathbf{P})^{-1} \\
&= \mathbf{P}^{-1}(\mathbf{I} - \widetilde{\mathbf{C}})^{-1}\mathbf{P}^{-\prime} \\
&= \mathbf{P}'(\mathbf{I} - \widetilde{\mathbf{C}})^{-1}\mathbf{P} ;
\end{aligned}
$$

where $(\mathbf{I} - \widetilde{\mathbf{C}})^{-1}$ admits a representation in terms of the geometric series (see Appendix B), which in this case consists in a sum of lower triangular matrices.

## Appendix D  Locally Stationary Wavelet Process

In the univariate case, *locally stationary wavelet* process (LSW, Nason et al. 2000) is a suitable modeling framework to represent a non-stationary process $\mathbf{x}_T$ of length $T = 2^J$, $J \in \mathbb{N}$, by means of a triangular

multiscale representation

$$x_T[t] = \sum_{j=1}^{J} \sum_{k=-\infty}^{+\infty} v_j[k/T] z_{j,k} \psi_j[t-k].$$

(8)

The building blocks of Equation (8) are: (i) the random amplitude $v_j[k/T] z_{j,k}$ composed by a time-varying amplitude $v_j[k/T]$ and a normal noise variable $z_{j,k}$ such that $cov(z_{j,k}, z_{j',k'}) = \widetilde{\delta}_{j,j'} \widetilde{\delta}_{k,k'}$, where $\widetilde{\delta}_{j,j'}$ represents the Kronecker delta; (ii) discrete, real valued and compactly-supported oscillatory functions $\psi_j[t-k]$, namely *non-decimated wavelets*. At each time only some values contribute to $x_T[t]$, and the time-dependence is managed by the index $k$. Local stationarity means that the statistical properties of the process vary slowly over time. This feature is essential in order to make learning possible (Nason et al., 2000). Within the LSW framework, local stationarity is formalized by means of a smoothness assumption concerning the time-varying amplitudes $v_j[k/T]$ (Fryzlewicz et al., 2003). Indeed, the latter quantity provides a measure of the time-dependent contribution to the variance at a certain time scale level $j \leq J$, namely the *evolutionary wavelet spectrum* (EWS), defined as $S_j[\nu] = |v_j[\nu]|^2$, with $\nu = k/T$ being the rescaled time (Dahlhaus, 1997). For a stationary process, EWS is constant $\forall j \leq J$. As an example, consider the $MA(1) = 1/\sqrt{2}(\epsilon[t] - \epsilon[t-1])$. We obtain it by setting in Equation (8) the following values for the previous components: (i) $z_{j,k} = \epsilon[t]$; (ii) $S_j = 1$ if $j = 1$ and zero otherwise; $\psi_1 = [1/\sqrt{2}, -1/\sqrt{2}]$ as the Haar wavelet. Because $S_1$ is constant and different from zero only for $j = 1$, we obtain a stationary amplitude $w_1[\nu] = 1$ only for the first scale level. Then, it follows that

$$\begin{aligned} x_T[t] &= \sum_{j=1}^{J} \sum_{k=-\infty}^{+\infty} v_j[k/T] z_{j,k} \psi_j[t-k] \\ &= \sum_{k=-\infty}^{+\infty} 1 \cdot \epsilon_{1,k} \psi_1[t-k] \\ &= \frac{1}{\sqrt{2}} (\epsilon[t] - \epsilon[t-1]). \end{aligned}$$

## Appendix E  Definitions of the Performance Metrics

In this section we describe the metrics used to evaluate the goodness of the estimated adjacency tensor of the causal graph and the retrieved causal ordering.

**Adjacency.** For the predicted adjacency tensor, we monitor both accuracy and structural scores.

With regards to accuracy measures, we look at the true positive rate ($TPR$, recall), the false discovery rate ($FDR$, 1-precision) and the F1-score. The first is defined as $TP/P$, where $TP$ is the number of predicted edges that exist in the ground truth with the same direction and $P$ (condition positive) is the number of links in the ground truth. The second given by $FP/(FP + TP)$. Here, $FP$ is the number of edges that do not exist in the skeleton of the ground truth, i.e., in the undirected adjacency. Finally, the F1-score is computed as the harmonic mean between $TPR$ and $1 - FDR$ (precision).

Concerning structural metrics, first we consider the *structural Hamming distance* ($SHD$), that represents the number of modifications (added, removed, reversed edges) needed to retrieve the ground truth starting from the estimated network. Then, we also monitor the ratio between the number of predicted edges and the condition positive, given by $NNZ/P$, where $NNZ$ represents the sum of directed ($D$) and undirected ($U$) estimated edges. Finally, we have the fraction of predicted undirected edges, computed as $FU = U/NNZ$.

**Causal Ordering.** To compare the estimated causal ordering with the ground truth, we consider three metrics able to provide a measure of the association strength between two rankings.

First, we look at Kendall-$\tau$, which is a measure of ordinal correspondence between two rankings, bounded between $-1$ (low correspondence) and 1 (strong correspondence). Given two orderings $\hat{\prec}$ and $\prec$, the statistics is defined as:

$$\text{Kendall-}\tau = (P - Q)/\sqrt{((P + Q + T) \cdot (P + Q + U))},$$

where here P is the number of concordant pairs, Q the number of discordant pairs, T the number of ties only in $\hat{\prec}$, and U the number of ties only in $\prec$;

Second, we employ a measure of ranking quality widely applied in information retrieval, the *normalized discounted cumulative gain* (nDCG). Consider a ground truth ordering $\prec$ of length $N$ and suppose to associate items with descending scores $s$, from $N$ to 1. Then, consider an other ordering $\hat{\prec}$ over the same set of elements in $\prec$. Now, define the *discounted cumulative gain* (DCG) as:

$$DCG = \sum_{i=1}^{N} \frac{s_i}{\log_2(i+1)} \, ,$$

and let the *ideal discounted cumulative gain* (IDCG) to be the DCG of $\prec$. Therefore, the nDCG is defined as the ratio by the DCG and the IDCG. This score is bounded between the nDCG of the worst ordering of scores $\bar{s}$, i.e., $s$ sorted in ascending order, and 1. In our analysis we use a min-max scaling to map nDCG to the unit interval. To evaluate the capability of a method in providing high-score items at first positions $k$, we compute the nDCG@k by considering only the first $k$ elements of $\hat{\prec}$.

Finally, we consider Spearman's rank correlation $\rho_S$, that provides a non-parametric correlation coefficient between two series. Here, differently from Pearson correlation, data is not assumed to be normally distributed. Thus, as Kendall-$\tau$, this statistics is bounded between $-1$ and 1. Since in our case we have two score vectors, namely $\hat{\prec}$ and $\prec$, made by distinct values, this metrics can be computed as:

$$\rho_S = 1 - \frac{6\sum_i(\hat{\prec}_i - \prec_i)}{N(N^2-1)} \, .$$

## Appendix F    Additional Monitored Metrics

In this section we provide additional analysis to better understand the behaviour of the considered methods when (i) we navigate the $(\tau, \mu)$-quadrant keeping the other parameters fixed, (ii) we vary the density of the MN-DAG at a point in the quadrant, and (iii) we change the size of the MN-DAG at a point in the quadrant. Figure 16a refers to the first experimental setting and shows that MN-CASTLE reduces the number of false discoveries returned by baseline models, especially when $\mu \neq 0$. On the contrary, in the single-scale stationary case, best $FDR$ values are provided by GOLEM. Figure 16b tells us that MN-CASTLE is the best in the retrieval of true positives in all cases. Moreover, the estimated to true edge set ratio given in Figure 16c shows that our model tends to slightly overestimate the number of causal connections, while baseline models tend to return sparser causal structures. Finally, we plot the fractions of undirected connections given by our model. Even though we do not impose any causal ordering during the evaluation phase, we see that our model rarely returns undirected edges.

With regards to causal ordering estimation, Figure 17 provides results also for Kendall-$\tau$ statistics. Here, the methodology used is the same as in Section 4.2. According to this metric, MN-CASTLE outperforms the baseline model in all cases. In addition, Kendall-$\tau$ values do not lower as $\tau$ grows and improve as $\mu$ increases.

Hence, Figures 18 and 19 depict the resulting values for the same metrics above, obtained in the second and third experimental settings, respectively. Overall, when the density of the network grows, the metrics improve for all methods. Moreover, MN-CASTLE provides the best performance for all values of $\delta$. When we vary the network size, our method still outperforms the baselines for $N = 5$ and $N = 10$. However, for the remaining two values of $N$, we observe a worsening of the performances. Notably, Figure 19b shows that the $TPR$ is greatly reduced compared with that of MSCASTLE, which shows no dependence on network size. Furthermore, looking at Figure 19c, we see that as $N$ increases the network estimated by MN-CASTLE becomes much sparser.

Finally, Figure 20 depicts the results for Kendall-$\tau$ statistics related to causal ordering estimation. Here, we see that MN-CASTLE performance grows along with $\delta$ and does not show dependence (in mean terms) on the size of the underlying MN-DAG.

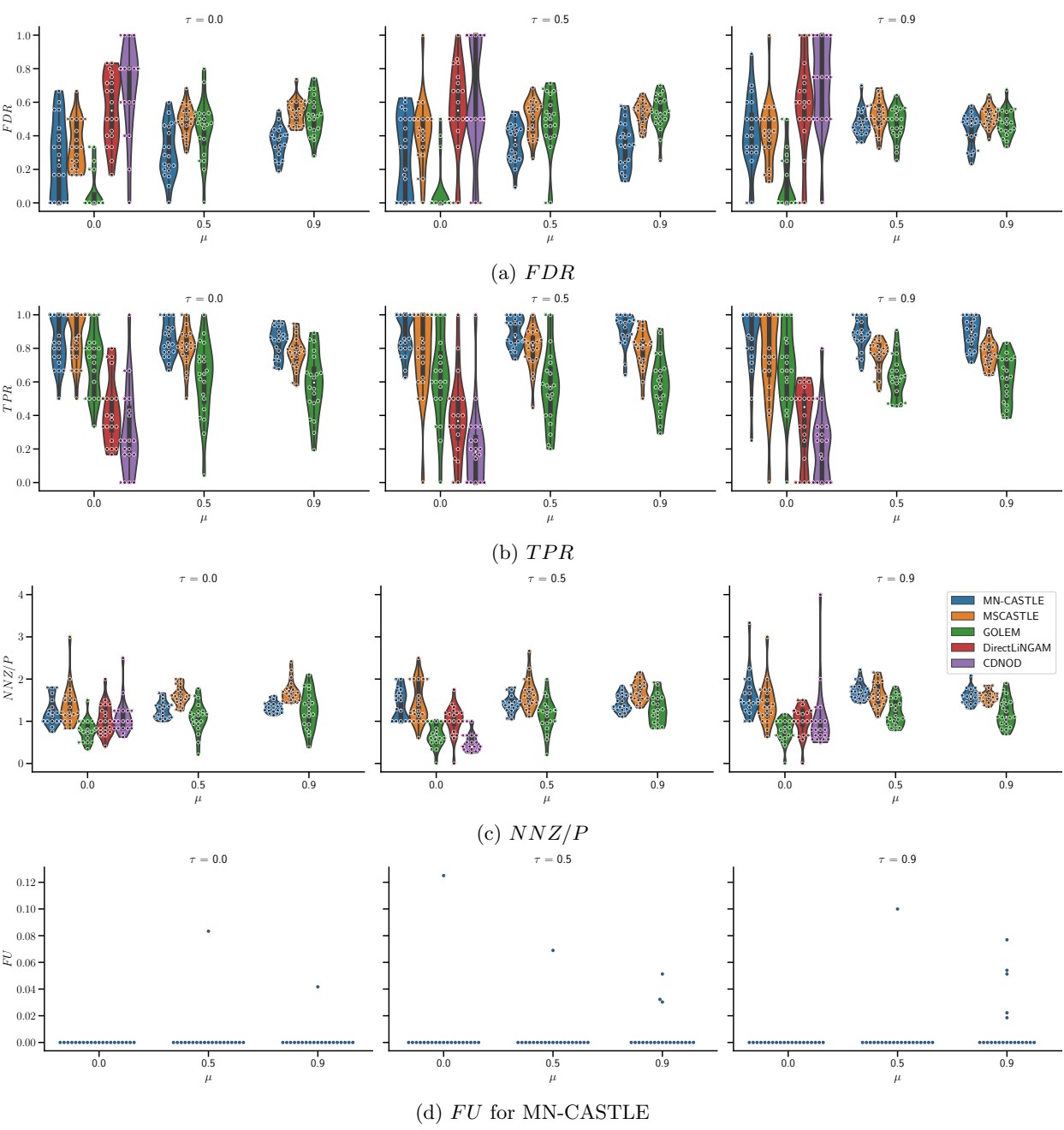

Figure 16: The figure depicts results returned by additional monitored metrics for the considered $(\tau, \mu)$ settings. Here we use violin plots, where we overlay a box-plot to visualize also IQR. In addition, we report the values of the metric, plotted as points.

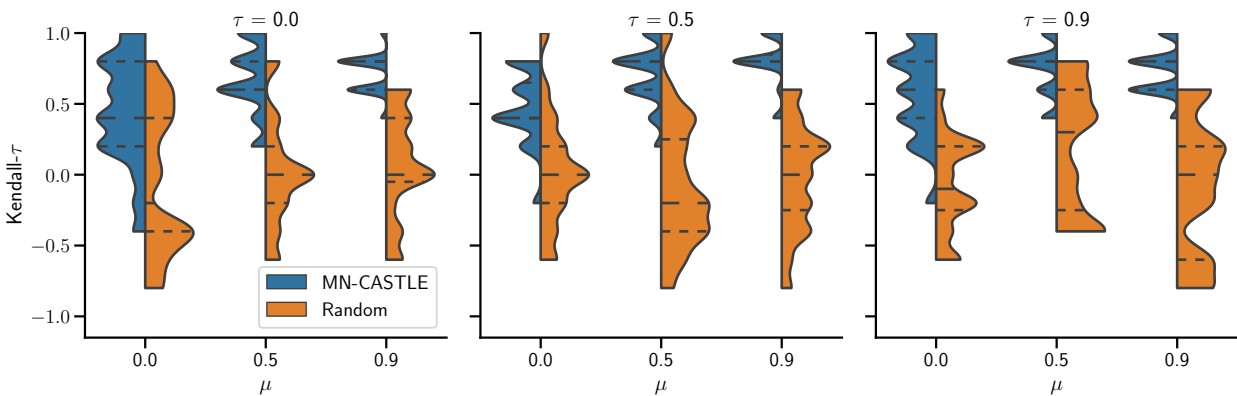

Figure 17: The figure depicts violin plots along with quartiles reference lines (dashed lines) for Kendall-$\tau$ metric. MN-CASTLE is given in blue while a random baseline model in orange. For every dataset generated according to a given $(\tau, \mu)$ setting (i) we sample $1 \times 10^3$ causal orderings $\hat{\prec} \sim PL(\hat{\boldsymbol{\theta}})$, where $\hat{\boldsymbol{\theta}}$ is the estimated vector of scores; (ii) we draw $1 \times 10^3$ random causal orderings $\bar{\prec} \sim PL(\bar{\boldsymbol{\theta}})$, where $\bar{\theta}_i \sim U(0, N)$, $i = 1, \ldots, N$. Afterwards, we evaluate the Kendall-$\tau$ by using the sampled causal orderings and $\prec$ for both models. Thus, each violin plot is made by $2 \times 10^4$ points.

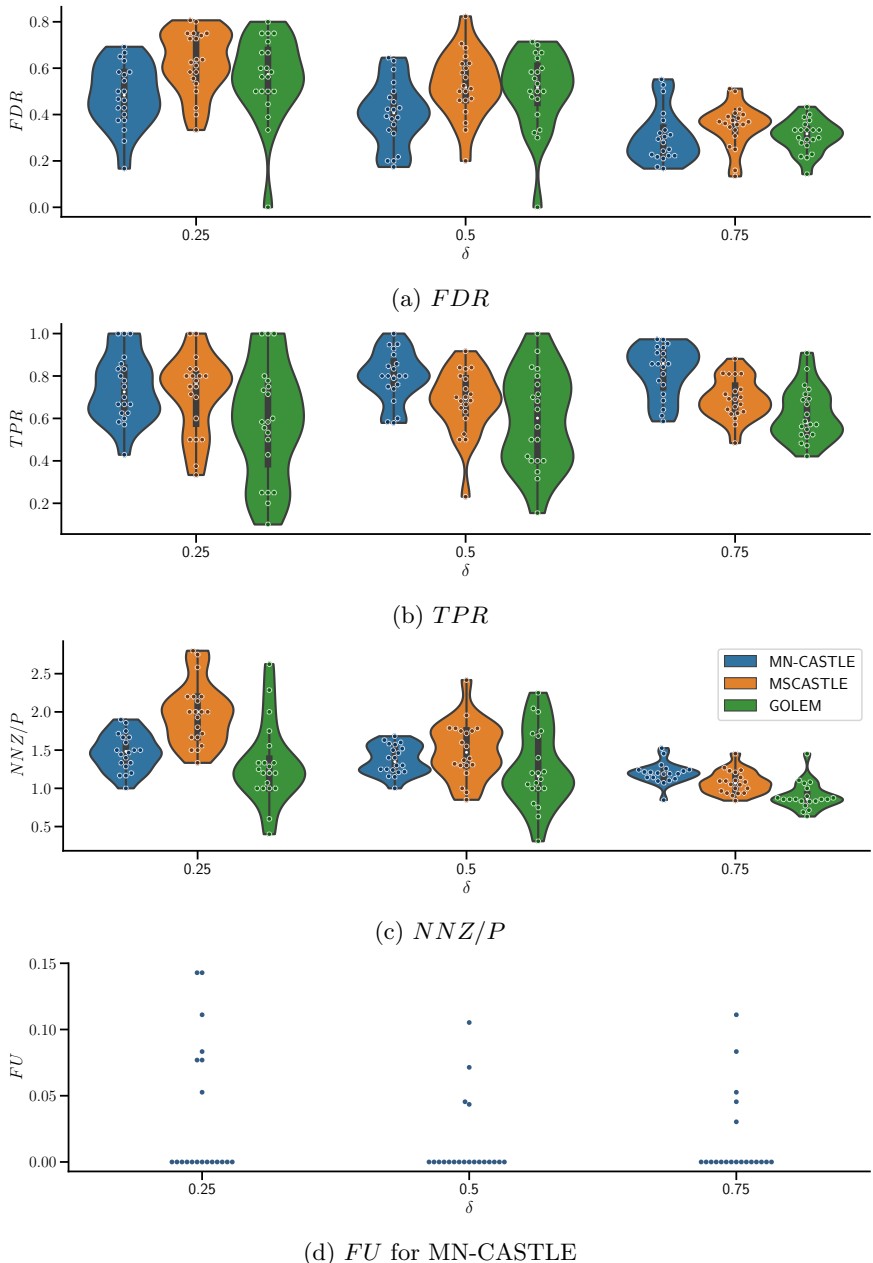

(a) $FDR$

(b) $TPR$

(c) $NNZ/P$

(d) $FU$ for MN-CASTLE

Figure 18: The figure depicts results returned by additional monitored metrics for different MN-DAG densities $\delta$. Here we use violin plots, where we overlay a box-plot to visualize also IQR. In addition, we report the values of the metric, plotted as points.

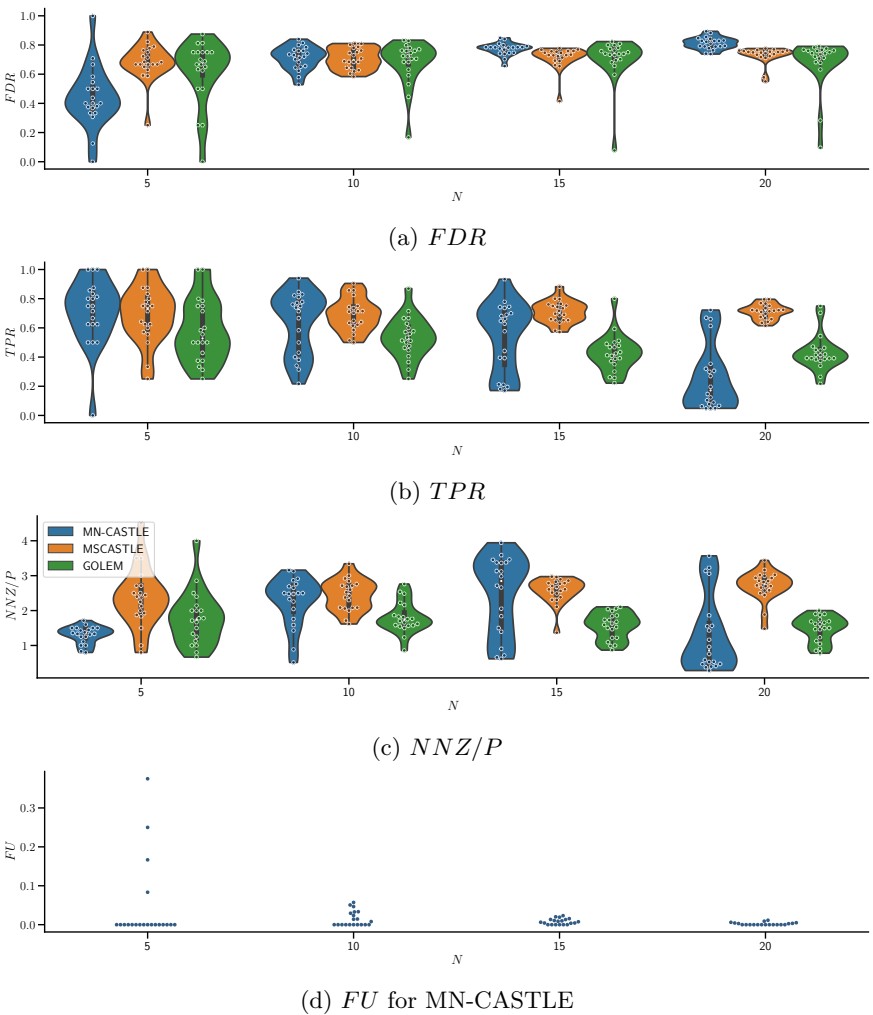

(a) $FDR$

(b) $TPR$

(c) $NNZ/P$

(d) $FU$ for MN-CASTLE

Figure 19: The figure depicts results returned by additional monitored metrics for different MN-DAG number of nodes $N$. Here we use violin plots, where we overlay a box-plot to visualize also IQR. In addition, we report the values of the metric, plotted as points.

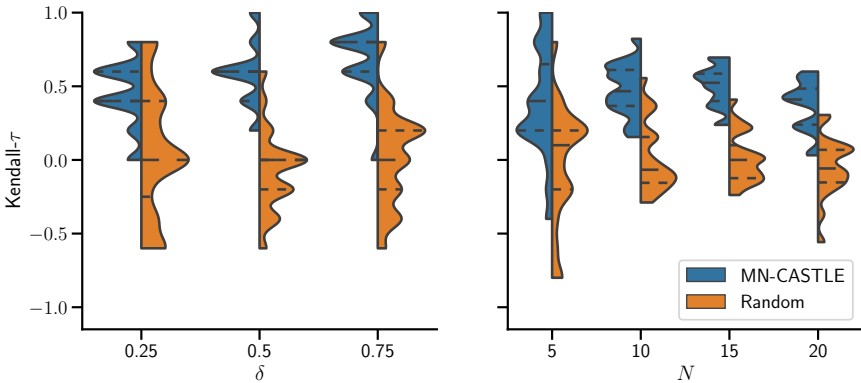

Figure 20: The figure depicts violin plots along with quartiles reference lines (dashed lines) for Kendall-$\tau$ metric. MN-CASTLE is given in blue while a random baseline model in orange. On the left, we vary the MN-DAG density $\delta$, whereas on the right we vary the number of nodes $N$, as described in Section 4.2. For every dataset generated according to a given setting (i) we sample $1 \times 10^3$ causal orderings $\hat{\prec} \sim PL(\hat{\boldsymbol{\theta}})$, where $\hat{\boldsymbol{\theta}}$ is the estimated vector of scores; (ii) we draw $1 \times 10^3$ random causal orderings $\bar{\prec} \sim PL(\bar{\boldsymbol{\theta}})$, where $\bar{\theta}_i \sim U(0, N)$, $i = 1, \ldots, N$. Afterwards, we evaluate the Kendall-$\tau$ by using the sampled causal orderings and $\prec$ for both models. Thus, each violin plot is made by $2 \times 10^4$ points.

## Appendix G   Models Configuration

Below we report the models hyper-parameters used during the test phase:

- MN-CASTLE: fraction of inducing points equal to 64%; $K = K_{\mathrm{RBF}}$; in case $\tau = 0$ (the estimated $\hat{S}_j$ is constant) we use as prior for $\lambda_K$ a normal $N(1. \times 10^3, 1. \times 10^{-3})$; number of iterations iter$= 6. \times 10^2$ with 10 particles;

- MSCASTLE: $\ell_1-$ penalty parameter $\lambda = 1. \times 10^{-1}$; pruning threshold $\gamma = 5. \times 10^{-2}$; Daubechies wavelet with filter length equal to 2; maximum value for dagness function $h_{\mathrm{tol}} = 1. \times 10^{-8}$;

- GOLEM: pruning threshold $\gamma = 5. \times 10^{-2}$; number of iterations iter$= 1. \times 10^4$;

- DirectLiNGAM: pruning threshold $\gamma = 5. \times 10^{-2}$;

- CDNOD: independence test = Fisher's Z; significance level $\alpha = 95\%$.

## Appendix H   Evolution over Time of Estimated Causal Relations

We apply MN-CASTLE over a synthetic dataset constituted by $N = 5$ time series of length $T = 100$ each. To generate the data, we use the exposed probabilistic model over MN-DAGs, where we set $\tau = \mu = \delta = 0.5$ and we use as $K = K_{\mathrm{RBF}}$. In this case, we obtain $J = 3$ scale levels. The configuration of MN-CASTLE is the same as Appendix G.

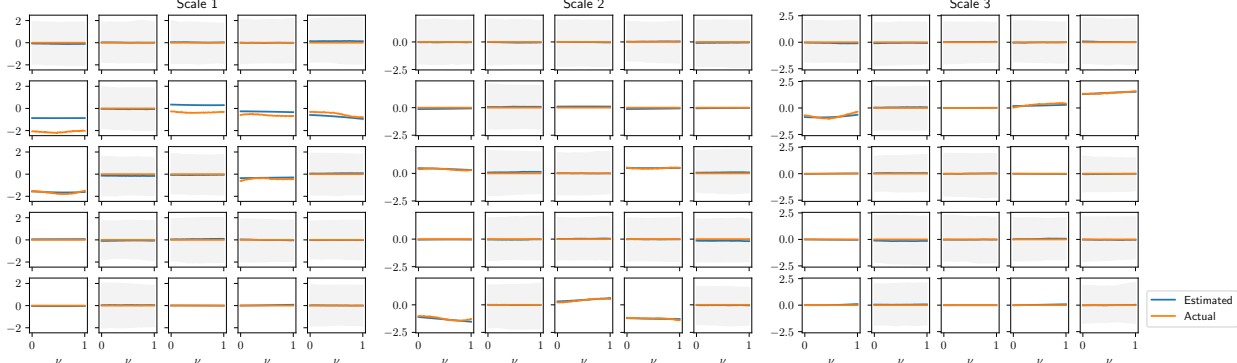

Figure 21: The figure depicts the evolution over time ($\nu = t/T$) of estimated causal coefficients (blue) vs the ground truth latent coefficients (orange), for the three temporal resolutions. Light blue bands refer to 99% confidence level while the red dashed line indicate the zero.

Figure 21 depicts the estimated causal relations and their evolution over time. MN-CASTLE correctly tracks the behaviour of latent causal coefficients in most cases. As given in Section 3.3, in the second inference step we use the mode of $PL$ distribution $\hat{\prec}^0$ to mask the distribution of hidden functions $\mathbf{f}$. Consequently, only those relations that conform to the experienced causal orderings show tight 99% CIs.

