# OpenReview forum: "Multiscale Non-stationary Causal Structure Learning from Time Series Data"
_TMLR — Rejected by TMLR_

### Review · Reviewer_9S9i · 2022-10-31

**Summary Of Contributions:**

The paper studies the problem of causal structure learning from time-series data. This paper proposes multiscale non-stationary directed acyclic graphs (MN-DAGs) which generalizes linear DAGs to the time-frequency domain. The authors then develop a probabilistic generative model over MN-DAGs, of which the latent variables correspond to the causal ordering and causal relationships. A Bayesian method (called MN-CASTLE) for learning the MN-DAGs is further proposed by using stochastic variational inference. Experiment results on synthetic and real world data are provided.

**Audience:**

Yes

**Claims And Evidence:**

Yes

**Requested Changes:**

In my opinion, the following changes are critical to the paper, each of which correspond to one of the weaknesses discussed above.
- Provide discussion about how the proposed formulation/method differs from existing causal structure learning methods that handle non-stationarity/heterogeneity or evolving causal structures, e.g., [1, 2, 3, 4]
- Provide discussion about the similarity and difference between several main components of the proposed method and existing causal structure learning works, like [5, 6, 7, 8]. Further empirical results would also strengthen the work; e.g., how does the proposed method of generating causal ordering empirically perform as compared to that in [5, 6]?
- Provide discussion/analysis about identifiability and consistency of the proposed formulation/method.

**Strengths And Weaknesses:**

Strength:
- The paper is well written and clear. The theoretical background is appropriately introduced before presenting the proposed method.
- The topic considered in this work is interesting and important because it is common to encounter non-stationary in real world.
-  Experiments are conducted across different settings and the results are convincing.

Weakness:
- Lack of discussion about how the proposed formulation/method differs from existing causal structure learning methods that handle non-stationarity/heterogeneity or evolving causal structures, such as Huang et al. (2020) and [1, 2, 3, 4].
- Several main components of the proposed method have been used in existing causal structure learning works and are not discussed/compared. In particular, similar strategy of sampling the causal ordering (via permutations) and variational inference have been adopted by [5, 6]. Also, the strategy of generating logical mask has been adopted by [7, 8].
- Lack of discussion/analysis about whether the proposed formulation is identifiable, and whether the proposed causal structure learning method is consistent.
- In Appendix A, based on my understanding, the step of writing $(I-C)^{-1}$ as an infinite series requires the condition that the spectral radius of $C$ is smaller than one. However, the proof did not mention anything about this aspect.

References:
[1] Perry et al. (2022). Causal Discovery in Heterogeneous Environments Under the Sparse Mechanism Shift Hypothesis.
[2] Ghassami et al. (2018). Multi-domain causal structure learning in linear systems.
[3] Strobl et al. (2019). Improved Causal Discovery from Longitudinal Data Using a Mixture of DAGs.
[4] Song et al. (2009). Time-varying dynamic Bayesian networks.
[5] Charpentier et al. (2022). Differentiable DAG Sampling.
[6] Cundy et al. (2021). BCD Nets: Scalable Variational Approaches for Bayesian Causal Discovery.
[7] Ng et al. (2022). Masked Gradient-Based Causal Structure Learning.
[8] Ke et al. (2020). Learning Neural Causal Models from Unknown Interventions

---

> ### Author Response · Authors · 2022-11-15
> **Reply to requested changes (1/3)**
>
> Dear reviewer, thanks for raising the above points. Please, find below our comments.
>
> 1.  Beyond the specific inference techniques used, the methods outlined in the above articles aim to infer causal structures in the presence of non-stationarity, under certain assumptions.
> The main (implicit) assumption common to all approaches, concerns the time scale at which causal interactions occur, that is, it is assumed that this scale coincides with the frequency of observation of the data. The model we propose relaxes this assumption, and allows time-dependent causal relationships to be investigated at different temporal resolutions.
> Another difference concerns the assumption regarding the existence of multiple domains. Here, causal dependencies between variables may vary across domains, but are assumed to be stationary within each of them. In this way it is possible to exploit non-stationarity and distributional shifts to recover the underlying causal structure. Although in the context of time series, the dataset can be segmented into different domains through a sliding window approach, this procedure introduces discretionary choices such as (i) the choice of the dataset's splitting points and (ii) the size of the time window in which causal relationships should be stationary. Indeed, in general, for real data there is no prior knowledge regarding the above issues: the causal structure might vary considerably even when windows are overlapping [9] (and methods such as IB and MC in [2] even require non-overlapping windows). Hence, our approach aims to learn the causal structure and describe its temporal evolution, assuming that it is linear in the frequency domain, and that the causal ordering is shared between the temporal resolutions considered. In addition, in our case the parents of each node at timestamp $t$ can vary over time, through the increase/decrease of causal relations, but always respecting the global causal ordering. This constitutes a departure from [1], in which the parent set of nodes is shared by each domain and from [3], where causal structures can also contain cycles.
> Finally, as in [4], our model is linear with causal coefficients evolving smoothly over time. However, the model proposed in [4] is autoregressive over time with lag equal to $1$. In contrast, building on the mathematical theory of multivariate locally stationary wavelet processes, our approach does not assume any lag, but rather defines $\mathbf{X}[t]$ as the sum of contributions from different time scales. In detail, the random amplitudes of each scale at a certain instant $t$ are determined by a mix of the scale-related latent noise vector. Hence, in our work the linearity of the (instantaneous) relations manifests itself in the frequency domain. Furthermore, although these relations are instantaneous, since they live on different temporal resolutions, they induce lagged auto/cross-correlation in the generated data (see [10]).

---

> ### Author Response · Authors · 2022-11-15
> **Reply to requested changes (2/3)**
>
> 2.  Our probabilistic generator extends both DP-DAG [5] and BCD Nets [6]. It is suitable for time-series data and provides a causal structure that lives in the time-frequency domain.
> Therefore, our proposed model has additional features compared to previous models, concerning i) the sampling of temporal resolutions (pages of the graph), ii) the decoupling of these with global causal ordering, and iii) a three-component structural equation that differentiates causal relationships across different temporal resolutions and over time.
> Then, two steps are shared by all procedures: the sampling of causal ordering and sparsity constraint. Our model and DP-DAG leverage Gumbel distributed variables for permutation vector sampling. Next, our model applies a Sort operator while DP-DAG an argmax of the SoftSort operator. BCD Nets, on the other hand, uses Gumbel-Sinkhorn distribution. In terms of causal ordering generation, our model has the lowest computational cost: $\mathcal{O}(N \log(N))$ vs $\mathcal{O}(N^2)$ and $\mathcal{O}(N^3)$ of DP-DAG and BCD Nets. Regarding sparsity, our generative model uses a Bernoulli distribution, while DP-DAG the argmax of a Gumbel-Softmax distributed variable. BCD Nets, on the other hand, uses a horseshoe prior to promote sparsity.
> In the learning phase, MN-CASTLE uses the REINFORCE estimator equipped with a data-dependent control variate strategy for learning the parameters of the Plackett-Luce distribution and uses masking to optimize only the causal relations compliant to the mean causal ordering of the Plackett-Luce distribution. Unlike [7] and [8], the masking used in our model does not eliminate all non-causal inputs, but rather eliminates all possible causal structures that do not conform to the imposed causal ordering. The reason for this choice lies in the fundamental observation that, given a causal ordering, feedback loops between variables are excluded and the problem is no longer NP-hard [11].
> Our model associates each causal relationship with a confidence interval. It is on the basis of the latter that the causal coefficient is judged to be statistically different from zero, not on the basis of a final pruning/thresholding as in previous work. For example, in Fig.14b the NKX $\rightarrow$ SHC and UKX $\rightarrow$ NKX interactions have mean values different from zero, yet the confidence intervals indicate that they are not statistically different from zero.
> Finally, we exploit recent developments in variational inference in order to approximate the posterior distribution over MN-DAG parameters given data, in accordance with the MN-CASTLE probabilistic model. This general learning scheme is also exploited in other recent works [12, 13] to model the posterior distribution over the parameters of a DAG, as defined in the corresponding proposed probabilistic models.

---

> ### Author Response · Authors · 2022-11-15
> **Reply to requested changes (3/3)**
>
>
> 3.  We included a brief discussion concerning identifiability in Sec.5. Identifiability is a crucial point and causal inference from observational data is a tough task also because of non-identifiability issues. Studying the assumptions that make the model described in Equation (4) identifiable is for sure an interesting research direction.
> Even though some identifiability results have been proved for some classes of linear structural equation models, under different types of restrictions [14,15,16,17], the case of MN-DAG needs to be carefully investigated due to the presence of non-decimated wavelet transform, the unobservability of the contributions to the process coming from each time resolution, and the linearity of the model in the frequency domain.
> 4.  Many thanks for raising the point concerning the geometric series representation of $(\mathbf{I}-\mathbf{C})^{-1}$. We specified in Appendix B that this is allowed by the fact that nilpotent matrices have eigenvalues equal to zero and then are convergent.
>
> References: [1] Perry et al. (2022). Causal Discovery in Heterogeneous Environments Under the Sparse Mechanism Shift Hypothesis. [2] Ghassami et al. (2018). Multi-domain causal structure learning in linear systems. [3] Strobl et al. (2019). Improved Causal Discovery from Longitudinal Data Using a Mixture of DAGs. [4] Song et al. (2009). Time-varying dynamic Bayesian networks. [5] Charpentier et al. (2022). Differentiable DAG Sampling. [6] Cundy et al. (2021). BCD Nets: Scalable Variational Approaches for Bayesian Causal Discovery. [7] Ng et al. (2022). Masked Gradient-Based Causal Structure Learning. [8] Ke et al. (2020). Learning Neural Causal Models from Unknown Interventions. [9] D'Acunto, Gabriele, et al. (2021) "The evolving causal structure of equity risk factors." [10] Park et al. (2014) “Estimating time-evolving partial coherence between signals via multivariate locally stationary wavelet processes.” [11] Buntine, Wray. (1991) "Theory refinement on Bayesian networks." [12] Annadani, Yashas, et al. (2021) "Variational causal networks: Approximate bayesian inference over causal structures." [13] Lorch, Lars, et al. (2022) "Amortized Inference for Causal Structure Learning." [14] Shimizu (2006) “A linear non-gaussian acyclic model for causal discovery.” [15] Peters and Bühlmann (2014). “Identifiability of gaussian structural equation models with equal error variances.” [16] Loh and Bühlmann (2016) ”High-dimensional learning of linear causal networks via inverse covariance estimation.” [17] Park and Kim. “Identifiability of gaussian linear structural equation models with homogeneous and heterogeneous error variances.”

---

### Review · Reviewer_BPv4 · 2022-11-05

**Summary Of Contributions:**

This paper proposes a causal discovery framework that can handle causal relationships at different temporal resolutions and non-stationary data. In particular, to characterize multi-scale non-stationary causal relationships, the authors leverage an existing mathematical modeling framework--multivariate locally stationary wavelet processes (MLSW), which is suitable to represent N zero-mean processes as a sum of contributions coming from different temporal resolutions. Accordingly, the authors develop a Bayesian method for the estimation of causal relationships. The main assumptions are that causal relationships are linear and the causal ordering is fixed across different time resolutions.



**Audience:**

Yes

**Broader Impact Concerns:**

The proposed method will be useful in analyzing causal relationships among time series, such as financial data, because in many cases, the causal influences may happen in different time scales and are nonstationary.

**Claims And Evidence:**

Yes

**Requested Changes:**

1. In experimental verification, consider a bit larger causal graphs and give the computational complexity.

2. Compare with the baseline which estimates multiscale stationary causal relationships [D’Acunto et al. (2022)].

3. Empirically show how different graph densities, temporal resolutions, and nonstationary levels affect the estimation accuracy.

**Strengths And Weaknesses:**

Strengths:
1. This paper considers causal relationships in different temporal resolutions, which is often the case in real-world data and is novel as far as I know.

2. The paper is well presented.

3. The leveraged mathematical modeling framework and estimation method are reasonable.

Weaknesses:
1. The experimental verification does not seem enough. For example, the authors only consider very small causal graphs (N<=7), and in the simulated experiments, N is only 5. How large graphs can the proposed method handle? I would like to suggest the authors consider a bit larger causal graphs and give the computational complexity.

2. The proposed method makes use of the wavelet-based framework to handle the multiscale, which suffers from limited joint time-frequency resolution, as also mentioned by the authors.

3. The proposed method only works for linear causal relationships, which is quite restricted.

---

> ### Author Response · Authors · 2022-11-15
> **Reply to requested changes**
>
> Dear reviewer, thanks for your comments. New experiments related to the points that you raised are available in Sec. 4.2.
>
> 1.  We studied the behaviour of our model, along with the baselines, for changing values of size $N$ of the underlying MN-DAG, while keeping fixed the number of observations $T$ (third experimental setting in Sec. 4.2). Fig.9 shows that the performance in the estimation of the causal adjacency tensor of all models worsens as $N$ increases, which is somewhat expected. Even though MN-CASTLE outperforms for low values of $N$, we observe a faster deterioration of its performance w.r.t. baselines along with $N$. In particular, for the largest value of $N$, as depicted in Fig.19b and Fig.19c, $TPR$ is much lower than previous settings since the network estimated by our method becomes very sparse.
> We believe this effect is due to batched GPs estimation. Indeed, the violin plots on the right of Fig.12 experimentally prove that the capability of MN-CASTLE in the retrieval of the causal ordering does not show any dependence on $N$.
> Finally, in Sec. 3.3 we remark that the main contribution to the computational complexity of our method comes from the approximate GP model, that has complexity $\mathcal{O}(\bar{T}^3)$, where $\bar{T}$ is the number of the inducing points $\bar{T} < T$.
> 2.  We added in all experiments MSCASTLE (D’Acunto et al., 2022) as well. As shown in the paper, MN-CASTLE compares favorably w.r.t. the latter method. Moreover, MSCASTLE turns out to be the best performing among baselines.
> 3.  We included in the manuscript results showing how MN-CASTLE reacts to changes in the density of the underlying MN-DAG. Concerning the causal adjacency matrix estimation, Fig.8 shows that our method outperforms the baselines in all cases and that its performance increases along with the value of the density parameter delta.
> Regarding the estimation of the causal ordering, the violin plots on the right of Fig.12 show that the monitored metrics improve and that the spread of the estimated kernel densities reduces as delta increases.
> Finally, Fig.13 demonstrates that the capability of MN-CASTLE in estimating the non-stationarity parameter $\tau$ does not depend on $\delta$.
> Figs.7, 10 and 11 depict the behavior of several performance metrics when we explore the $(\tau, \mu)$-quadrant.

---

### Review · Reviewer_dycz · 2022-11-08

**Summary Of Contributions:**

The paper proposes MN-DAG, which extends structural causal models to include nonstationarity and a multiscale property (causal relations changing over time).

A generative model is describe which involves (i) sampling the number of time scales from a binomial distribution, (ii) sampling a causal ordering according to the Placket-Luce distribution for each time scale, (iii) sampling and masking to generate weight matrices. Data may then be sampled from this model using the locally stationary wavelet process.

The authors then describe MN-CASTLE, an algorithm for learning MN-DAGs from such data using stochastic variational inference.

Experiments show data sampled from the MN-DAGs have the expected properties and MN-CASTLE is more effective at learning from synthetic data simulated from using the described procedure when the multiscale property is high.

**Audience:**

Yes

**Claims And Evidence:**

No

**Requested Changes:**

Background and paper clarity
- 'Multiscale' is never explicitly defined in the paper nor is the specific definition of stationarity that is being used. This can be gleaned from later parts of the text, but these should be defined explicitly since they are the basic for the proposed model
- The authors divide approaches for learning causal structures into 3 categories (i) constraint-based methods (ii) score-based methods and (iii) 'structural causal models' which includes approaches like LiNGAM and ANM, but this is not quite correct. The term 'structural causal model' has been used in the literature at least as far back as Pearl, 2000 to refer to the causal inference framework used in this paper (and to distinguish it from the potential outcomes framework), not the learning algorithm. Using 'structural causal models' to refer to a category of learning algorithms may lead to confusion.
- The authors should also consider reorganizing the paper so that the technical content in section 3 is described only as it is needed in section 4 once it has already been motivated.
- The authors should also consider carefully going through the introduction and removing details that the reader would need to have finished the rest of the paper to understand

Novel content
- The MN-CASTLE section reads as if it is an off the shelf application of SVI. If this is not the case, clarifying the novel parts might improve the paper
- Similarly, it is also unclear whether any of the sampling methods used are new or all were previously used in other approaches

Empirical results
- While MN-CASTLE applied to the results simulated directly from the generative model argue favorable for the approach when the multiscale property is high, this seems to be an overly pristine lab experiment given the level of complexity of the model and may not be realistic. The paper would be significantly improved if more experiments were provided where some assumptions are violated, e.g. nonlinearity, overlapping pages, etc.
- The real world results should have comparisons to other methods, especially since only one graph is provided. Otherwise it is not clear MN-CASTLE produces different results than existing, less complex, approaches.
- Since the experiments indicate the performance gains of MN-CASTLE come when the multiscale property is high, it would be helpful to see how the graph changes across pages for the real world example

**Strengths And Weaknesses:**

Strengths
- The model is ambitious in permitting both nonstationarity and the multiscale property
- The multiscale property, to my knowledge, has not been considered in previous work
- Extensive background on the building blocks that the method uses are included

Weaknesses
- The structure of the paper makes it difficult to read: the introduction delves too much into technical details using terms that are not yet defined; methods are defined before they are needed and before they have been motivated by the problem setup
- Some background on causal inference methods is not quite correct (see below)
- The method itself is limited in novel content; both the generative model and learning procedure are based on off the shelf methods and frameworks
- Given the limited novel contributions, the experimental results are not very extensive: while they show increased performance when data are simulated exactly according to the generative model and have significant multiscale, this may not be very realistic; the only real world data example is not compared to existing methods

---

> ### Author Response · Authors · 2022-11-15
> **Reply to requested changes**
>
> Dear reviewer, we thank you for your suggestion. Please, find below the answers to your points.
>
> ## Background and paper clarity
>
> 1.  We defined the key concepts for the proposed model, i.e, multiscale, temporal resolutions, and causal ordering, in Sec.1 of the revised version of our paper.
> 2.  Many thanks for raising this point, we replaced ‘structural causal models’ with ‘structural equation models’ in the newly uploaded version.
> 3.  Thanks for this suggestion. We reorganized our work as indicated in the post announcing the new version of the manuscript.
> 4.  We modified the introduction in order to make it more understandable. We removed the technical content that requires notions introduced in subsequent sections of the paper to be comprehended. Finally, we included a roadmap to improve the readability.
>
> ## Novel content
> 1.  SVI represents a general learning scheme to model the posterior distribution over the parameters given data, which is also exploited in other recent work [1, 2, 3, 4]. With reference to MN-CASTLE, the novelty concerns the structure of the probabilistic model, that is i) the chosen distributions for inferring MN-DAG parameters, ii) the usage of approximated batched Gaussian processes as latent variable of the probabilistic model, associated with causal coefficients, iii) the overall alternating 2-steps inference procedure.
> In order to highlight the novel contribution of the paper we added a Related Work section and we moved to supplementary material part of the technical background.
> 2.  With regards to the distributions used for sampling, we exploit already existing methodologies.
>
> ## Empirical results
> 1.  Thanks for your comment. We provide additional experiments in Sec. 4.2, that further strengthen the performance of MN-CASTLE. Nevertheless, as reported in Sec. 5, we think that the development of nonlinear multiscale causal structure learning generative models and related inference methodologies are important to extend our work, and we leave it as future research direction.
> 2.  We provide the causal networks estimated by MSCASTLE and GOLEM in Sec. 4.3.
>    The network inferred by our method is more similar to that of MSCASTLE than that estimated by GOLEM, as indicated by the Jaccard score values in Table 1.
> Overall, the multiscale methods MN-CASTLE and MSCASTLE are able to identify more connections between the stock markets considered.
> In addition, MN-CASTLE also provides information regarding the evolution of these connections, pointing out their statistical significance (e.g., HSI $\rightarrow$ SHC, SHC $\rightarrow$ UKX in Fig.14b), their weakening/strengthening and possible sign changes over the time period analyzed.
> These issues may impair other methods, which assume causal relationships to be stationary, in recovering some connections.
> Furthermore, Figs. 15a and 15b also demonstrate that the mean causal ordering estimated by our method is congruent with the baselines, in most cases.
> Since MN-CASTLE allows us to inspect the scores associated with each stock market (i.e., the estimated vector of parameters for causal ordering distribution), we can draw conclusions concerning the confidence of our model in the positioning of the various markets within the causal order. For instance, Fig.14a shows that the scores within $\theta$ associated with UKX and NKX equity markets are very close. This underscores the uncertainty of our model regarding the positioning of those markets within the causal ordering, which is also supported by the presence of a statistically significant relation from NKX $\rightarrow$ UKX in Fig.14b.
> 3.  Thanks for the comment. Unfortunately, the only statistical significant temporal resolution we find is the one associated with scale level $j=1$. Accordingly, in the real world use case we restrict our attention to the only statistically significant time scale. As reported in Sec. 5, we believe this is justified by the fact that the period analyzed is characterized by financial turbulence, due to the outbreak of the pandemic. During this crisis, investors had to react quickly to the shocks that followed, generating sudden swings in stock prices at a finer time scale.
>
> [1] Charpentier et al. (2022). Differentiable DAG Sampling. [2] Cundy et al. (2021). BCD Nets: Scalable Variational Approaches for Bayesian Causal Discovery. [3] Annadani, Yashas, et al. (2021) "Variational causal networks: Approximate bayesian inference over causal structures." [4] Lorch, Lars, et al. (2022) "Amortized Inference for Causal Structure Learning."

---

### Author Response · Authors · 2022-11-15
**Revised version available**

Dear editor and reviewers, thank you for your insightful comments that helped us greatly improve our work. We uploaded a revised version of our manuscript, addressing all the issues raised in the reviews.

We next provide a summary of the main modifications done in the revised manuscript.

## Structure

We reorganized our paper as suggested by the reviewers. In particular:
-   Sec. 2 has been turned into a subsection of Sec. 1;
-   We added a section of related work, right after Sec.1, to properly collocate our work in the literature;
-   We removed the section “Theoretical Background”. The parts related to the SEM and MLSW frameworks have been integrated within Sec. 3.2. Here, the material concerning univariate LSW process has been moved to Appendix E, whereas that concerning SVI is available in Appendix A.

Hence, the structure of the main corpus of our manuscript is the following:

1.  __Introduction__. It highlights the importance of causal structure learning from observational data and points out multiscale and non-stationarity issues. Here, we also describe at a high level the contribution of our work (some details have been either omitted or described in other parts of the paper, when necessary). We first deal with the probabilistic generative model and then the devised inference model, termed MN-CASTLE. Finally, we provide a roadmap for the reader.
2.  __Related Work__. This section relates our method to existing ones, highlighting differences and similarities;
3.  __Methods__. It contains the novel methodological content of the paper. It is constituted by three subsections that show (i) how we sample MN-DAG (Sec. 3.1); (ii) how we generate data from MN-DAG (Sec. 3.2); (iii) how we propose to infer MN-DAGs from data (Sec. 3.3);
4.  __Results__. Here we provide three subsections:
	1.  Sec. 4.1 statistically describes data generated by the probabilistic generative 	model;
	2.  Sec. 4.2 regards tests on synthetic dataset. It provides details concerning the experimental settings; introduces baseline models; and deals with the results obtained;
	3.  Sec. 4.3 analyses a real world use case on global equity markets.
5.  __Conclusions and Future Research Directions__. This section concludes by providing additional discussion concerning our findings, outlines open questions and future research directions.


## Experiments
We enriched the experimental part of the manuscript. In particular:
-   __Synthetic data__. We provide results concerning two additional experimental settings. The first aims to study the effects of changes in the underlying MN-DAG density on MN-CASTLE performances. The second is intended to assess the capability of our inference method when the network size grows and the number of available observations remains fixed. A detailed description of these settings is given in Sec. 4.2, that also reports the obtained results. Here, we included MSCASTLE (D’Acunto et al., 2022) as additional baseline;
- __Real world use case__. -   We added and compared the networks inferred from real data by MSCASTLE and GOLEM to that estimated by MN-CASTLE (Sec. 4.3). We obtained that the network inferred by our method is more similar to that of MSCASTLE than that estimated by GOLEM. We also appreciate that both MN-CASTLE and MSCASTLE are able to identify more connections between the considered equity markets.
Moreover, since our method can handle evolving causal connections, it also retrieves relationships that are active for short time periods and that change their sign. These issues may impair MSCASTLE and GOLEM, which assume causal relationships to be stationary, in recovering some of those connections. As far as the causal ordering is concerned, MSCASTLE and GOLEM agree with MN-CASTLE in most cases.

---

### Decision · Action_Editors · 2023-01-11

**Recommendation:** Reject

**Comment:**

While the model proposed is interesting, I agree with reviewers BPv4 and dycz that the experiments do not sufficiently demonstrate the benefit and applicability of the proposed method:

* Per dycz’s review, I strongly encourage you to look at performance under model misspecification. As we know, all models are wrong (but some are useful). While I don’t think it is necessary to incorporate eg nonlinearities, I *do* think it is necessary to explore how well MN-CASTLE recovers structure if the data is *not* generated according to the generative process.
* In your real data example, you find only one lengthscale to be significant. Having the only real-world example be on data without (detectable) multi-lengthscale behavior is not the best advertisement for your approach. If multi-scale behavior is indeed a common feature of causal relations, I would expect you to be able to find an example that demonstrates such behavior.
* As noted in your response to BPv4’s comments about computational complexity and in Fig 9, it appears that MN-CASTLE struggles when faced with larger N. This is something that should really be explored in greater detail (and ideally rectified---you hypothesize that the effect is due to the batched GPs, which could be tested using alternative formulations, albeit at potentially greater computational cost).

Reviewer dycz also note a lack of novelty---the components of the model and algorithm are mostly pre-existing. While I see their point, I do not think lack of novelty of the components is a reason to reject *per se*. However, it does reinforce the need for strong empirical evidence of performance, which unfortunately is lacking in the current version.

**Audience:**

Yes

**Claims And Evidence:**

The methodology is sound, however the experimental results do not do a sufficient job of motivating the use of the MN-CASTLE